# TabICL: A Tabular Foundation Model for In-Context Learning on Large Data

**Jingang Qu** [1]  **David Holzmüller** [2][3]  **Gaël Varoquaux** [1]  **Marine Le Morvan** [1]

## Abstract

The long-standing dominance of gradient-boosted decision trees on tabular data is currently challenged by tabular foundation models using In-Context Learning (ICL): setting the training data as context for the test data and predicting in a single forward pass without parameter updates. While TabPFNv2 foundation model excels on tables with up to 10K samples, its alternating column- and row-wise attentions make handling large training sets computationally prohibitive. So, can ICL be effectively scaled and deliver a benefit for larger tables? We introduce TabICL, a tabular foundation model for classification, pretrained on synthetic datasets with up to 60K samples and capable of handling 500K samples on affordable resources. This is enabled by a novel two-stage architecture: a column-then-row attention mechanism to build fixed-dimensional embeddings of rows, followed by a transformer for efficient ICL. Across 200 classification datasets from the TALENT benchmark, TabICL is on par with TabPFNv2 while being systematically faster (up to 10 times), and significantly outperforms all other approaches. On 53 datasets with over 10K samples, TabICL surpasses both TabPFNv2 and CatBoost, demonstrating the potential of ICL for large data. Pretraining code, inference code, and pre-trained models are available at https://github.com/soda-inria/tabicl.

## 1. Introduction

Tabular data, structured in rows and columns, is widely used in industries like healthcare (Cartella et al., 2021) and finance (Johnson et al., 2016), where tabular classification problems underpin numerous real-world applications.

In other data modalities, foundation models – particularly Large Language Models (LLMs) (Zhou et al., 2024) – have significantly advanced the ability to tackle new tasks and few-shot learning. This is largely due to their remarkable in-context learning (ICL) capabilities (Brown et al., 2020), which enable them to capture patterns in prompts without requiring parameter updates. This success combined with the pervasiveness of tables have spurred interest in tabular foundation models (van Breugel & van der Schaar, 2024).

While LLMs are primarily designed to model natural language, attempts to fine-tune them for tabular data have emerged (Hegselmann et al., 2023; Fang et al., 2024, and references therein). These efforts rely on table serialization, which is the process of converting table rows into text or sentences which can then be tokenized. Gardner et al. (2024) fine-tuned Llama 3-8B on a corpus of serialized tables and demonstrated the effectiveness of this approach compared to tree-based models in the few-shot setting. However, such language models-based approaches are limited by the size of their context window to fit large serialized tables (e.g. up to 32 or 64 shots in Gardner et al. 2024). Moreover, it is unclear whether LLMs can effectively handle numerical values (Thawani et al., 2021). Finally, as evidence shows that LLMs are pretrained on many popular datasets (Bordt et al., 2024), their evaluation on tabular prediction tasks should also be conducted with care.

Taking a markedly different approach, Hollmann et al. (2022) introduced TabPFN, a transformer-based tabular foundation model for classification tasks, pretrained on synthetic tabular datasets only. A key feature of TabPFN is ICL with tables, which eliminates the need for tokenization and enables efficient handling of small tables with up to 1K samples and 100 features. The same authors then introduced TabPFNv2 (Hollmann et al., 2025), an improved version that significantly outperforms tree-based and neural network competitors on small-to-medium datasets with up to 10K samples and 500 features. The great promise of tabular ICL has spurred a new line of research (see Section 2.3), yet the quadratic cost of self-attention is a threat to the scalability of all of these. TabPFNv2 uses a two-way attention mechanism alternating between column-wise and row-wise attentions, which limits its scalability for large datasets. In real-world scenarios, where industrial datasets can contain millions of samples (Rubachev et al., 2024), the high computational

[1]SODA team, INRIA Saclay, France [2]Sierra team, INRIA Paris, France [3]Ecole Normale Supérieure, PSL Research University, Paris, France. Correspondence to: Jingang Qu <jingang.qu@inria.fr>.

*Proceedings of the 42nd International Conference on Machine Learning*, Vancouver, Canada. PMLR 267, 2025. Copyright 2025 by the author(s).

and memory demands of TabPFNv2 hinder its practicality.

In this paper, we introduce TabICL, a scalable and efficient tabular foundation model based on PFNs (Müller et al., 2024) and designed for classification tasks. Pretrained on synthetic datasets up to 60K samples, TabICL can effectively handle datasets up to 500K samples and 500 features, significantly expanding the scalability of ICL for tables.

To accommodate tables of arbitrary sizes, TabICL treats individual cells as fundamental units: each column is regarded as a set of cell values that capture feature-specific distributions and semantics, while each row comprises interdependent feature values. TabICL employs a two-stage architecture to achieve efficient ICL for tabular data. First, it encodes rows (excluding target labels) into dense vector embeddings. Each embedding is designed to capture the entire table information. This stage effectively collapses the column dimension to substantially reduce computational complexity and memory footprint for subsequent ICL. Second, it combines these compact yet informative embeddings with corresponding labels and then performs ICL. Therefore, the core of TabICL lies in its embedding strategy of the first stage, which is supposed to transform rows into semantically rich embeddings.

In natural language, words often carry clear semantics and are thus naturally associated to informative embeddings (Mikolov et al., 2013). However, tabular data lacks such an inherent structure: cell values can be ambiguous without metadata such as column names or data types. To address this challenge, TabICL adopts a well-constrained embedding strategy combining (1) distribution-aware column-wise embedding to capture statistical regularities within each column and (2) attention-based row-wise interaction to model dependencies across columns, thereby constructing semantically grounded representations for tabular data.

Feature embedding involves mapping scalar cell values into high-dimensional vectors for each feature, serving as a critical factor in model performance (Gorishniy et al., 2022). Since features often exhibit vastly different distributions, previous approaches typically use feature-specific embedding modules without parameter sharing, which, however, limits cross-table transferability. In this work, we reformulate feature embedding as a *set-input* problem, where a permutation-invariant set of cell values acts as input, and the output comprises corresponding one-to-one embeddings. To achieve this, we leverage the Set Transformer (Lee et al., 2019), a model specifically designed to process sets through efficient induced self-attention. It excels in tasks such as identifying extrema and counting unique elements, enabling the discovery of distribution-related metadata within each column and enhancing the ability to distinguish between features of different data types.

The feature embeddings are then processed per-row by another transformer, and aggregated into a single vector using learnable [CLS] tokens. This effectively captures complex feature interactions and accommodates a varying number of features. Overall, this column-then-row attention-based embedding achieves efficient sparse attention across all cells by leveraging the column/row inherent structure of tabular data as a strong inductive bias. Finally, the resulting row-wise embeddings are handled by a final transformer for ICL.

In addition to the above innovations, we introduce further improvements: (1) We refine TabPFN's pretraining synthetic datasets by adding a new tree-based data-generating model to incorporate the inductive biases of tree-based models (Grinsztajn et al., 2022; den Breejen et al., 2024; Grinsztajn, 2024); (2) We adopt curriculum learning by scaling the pretraining dataset size from 1K to 60K; (3) To address classification problems with over 10 classes (the pretraining limit), we use hierarchical classification (Silla & Freitas, 2011), breaking them into a hierarchical structure of sub-problems with $\leq 10$ classes. The increase in the number of tasks is largely offset by the fast predictions of TabICL.

To summarize our contributions: (1) We present TabICL, a novel scalable tabular foundation model for classification tasks that can accommodate any number of samples, features, and classes. In practice, TabICL handles up to 500K samples and 500 features with around 20GB of GPU memory; (2) We introduce a distribution-aware feature embedding approach that handles features with diverse properties in a unified manner, unlocking new possibilities for cross-table transferability; (3) TabICL performs tasks in a single forward pass and is orders of magnitude faster than tabular methods requiring hyperparameter tuning while still providing better performance in most cases. TabICL is also consistently faster than TabPFNv2 (up to 10 times), with efficiency gains increasing as dataset size grows; (4) We evaluate TabICL on the TALENT benchmark (Ye et al., 2025), comprising 200 classification datasets across various domains and sizes (up to 150K samples). TabICL performs comparably to TabPFNv2 on medium-sized datasets and significantly outperforms all other methods. On the 53 large datasets with over 10K samples, TabICL surpasses both TabPFNv2 and CatBoost (Dorogush et al., 2018). These results demonstrate the potential of ICL for large data.

## 2. Related Work

### 2.1. Foundation Models and In-Context Learning

In recent years, deep learning has been transformed by the emergence of foundation models, which are pretrained on massive, diverse datasets and serve as versatile backbones for downstream tasks. These transformer-based models enable In-Context Learning (ICL): performing tasks by analyz-

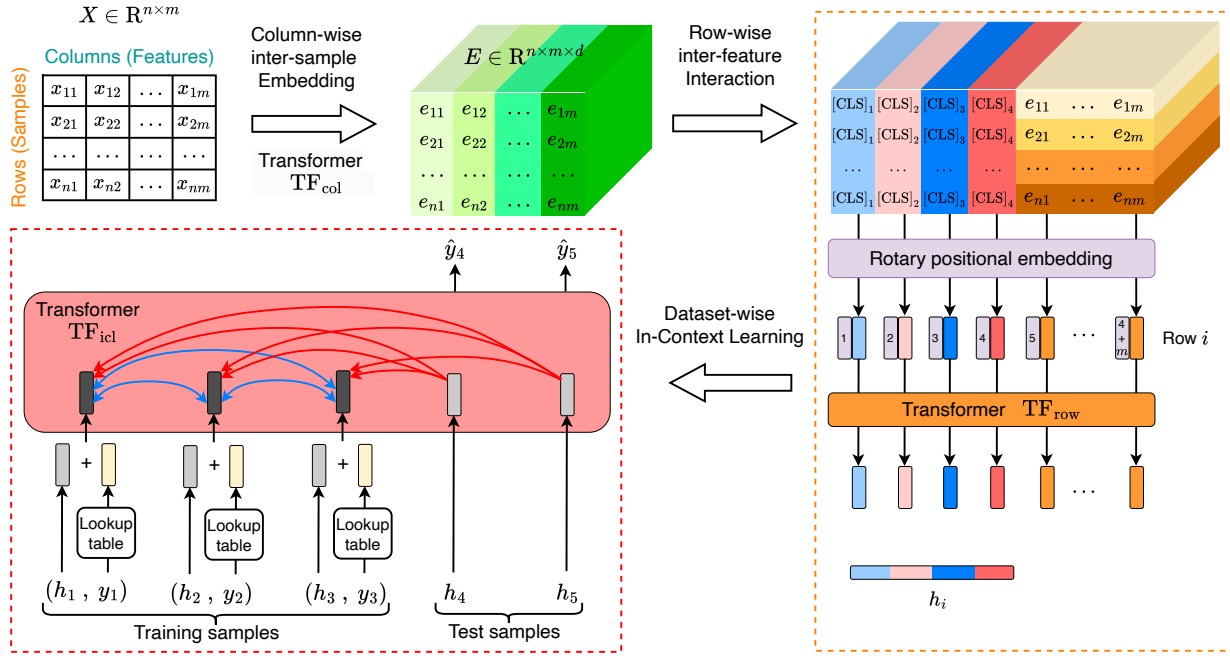

*Figure 1.* **Architecture Overview of TabICL.** First, column-wise embedding transforms each cell of the input table into an embedding vector using a transformer $TF_{col}$ (Section 3.2), producing $E$. Next, row-wise interaction prepends four trainable [CLS] tokens to $E$, applies rotary positional encoding, and processes $E$ row-by-row with a transformer $TF_{row}$. Concatenating the outputs of [CLS] tokens yields the final row-wise embeddings $H$. Finally, dataset-wise ICL operates on $H$ and uses a Transformer $TF_{icl}$ to predict the target labels for the test set in a single forward pass. Overall, TabICL consists of three transformers.

ing prompts containing input-output pairs, without explicit training or parameter updates. ICL operates as a form of on-the-fly reasoning. The mechanism underlying ICL remains elusive, with prevailing explanations framing it as implicit Bayesian inference (Xie et al., 2022; Müller et al., 2024), gradient descent optimization (von Oswald et al., 2023), and algorithmic learning (Garg et al., 2023).

### 2.2. Tabular Deep Learning Models

Gradient-boosted decision trees (GBDTs), such as CatBoost and XGBoost (Chen & Guestrin, 2016), have long dominated the tabular domain. However, growing efforts are focused on developing deep learning models for tabular data. Recent studies indicate a narrowing performance gap between GBDTs and tabular deep learning models (Ye et al., 2024; Gorishniy et al., 2024).

As tabular deep learning improves, cross-table transferability emerges as an important topic. Notable efforts in this direction include XTab (Zhu et al., 2023) and CARTE (Kim et al., 2024), which incorporate transferable components that are typically shareable backbone networks and dataset-specific components that require fine-tuning for each new task. The advent of tabular foundation models can bring new possibilities to cross-table learning, paving the way for large-scale pretraining and transfer learning across tables.

### 2.3. Tabular Foundation Models

TabPFN, short for Tabular Prior-Data Fitted Network, is a tabular foundation model. It is a transformer pretrained on extensive synthetic datasets to perform tabular classification tasks through ICL. TabPFN interprets ICL from a Bayesian perspective as an approximate posterior predictive distribution over synthetic datasets. Several variants aim to enhance its scalability, including distilling training data into a compact learned context via prompt tuning (Ma et al., 2024b; Feuer et al., 2024), selecting the most relevant subset of training data for each test sample (Xu et al., 2024; Thomas et al., 2024; Koshil et al., 2024), replacing quadratic with linear attention (Zeng et al., 2024), and generating small task-specific neural networks via an ICL-based hypernetwork (Müller et al., 2023). However, most variants do not structurally improve TabPFN but instead act as prompt engineering to reduce in-context samples.

Several TabPFN variants try to improve the quality of pretraining data, such as TabForestPFN (den Breejen et al., 2024) incorporating tree-based synthetic datasets and Tab-DPT (Ma et al., 2024a) using real-world datasets.

TabPFNv2 largely improves TabPFN in terms of both prediction performance and scalability. Our new model, TabICL, achieves comparable performance to TabPFNv2 while being

more scalable and computationally efficient. Appendix A provides a systematic comparison between TabPFNv2 and TabICL across architecture, pretraining, and scalability.

## 3. The TabICL Architecture

We consider a tabular classification task with an input space $\mathcal{X} \in \mathbb{R}^m$ and a target space $\mathcal{Y} \in [1, \cdots, C]$. Given a training dataset of input-output pairs $\mathcal{D}_{\text{train}} = \{(x_{\text{train}}^{(i)}, y_{\text{train}}^{(i)})\}_{i=1}^{n_{\text{tr}}}$ and test sample $x_{\text{test}}$, our goal is to predict the class probabilities $p(\cdot|x_{\text{test}}, \mathcal{D}_{\text{train}})$.

### 3.1. High-level Structure: Embedding then ICL

TabICL comprises two key modules: a tabular embedding module followed by an ICL module, with labels utilized exclusively in the ICL module. The tabular embedding module encodes table rows into dense vector representations while taking the inherent column-row structure into account. This module consists of two components: distribution-aware column-wise embedding, which captures statistical characteristics of individual columns, and context-aware row-wise interaction, which models dependencies between features. The ICL module subsequently processes both training and test embeddings through a transformer, enabling the prediction of the entire test set in a single forward pass through ICL. The overall architecture is depicted in Figure 1.

### 3.2. Distribution-aware Column-wise Embedding

The column-wise embedding (or feature embedding) maps each scalar cell in a column $c_j \in \mathbb{R}^n$ into a $d$-dimensional embedding. Unlike typical approaches that assign a separate embedding module to each column (Gorishniy et al., 2022), we embed all columns through a shareable Set Transformer $\text{TF}_{\text{col}}$, as illustrated in Figure 2b and formulated as follows:

$$W, B = \text{TF}_{\text{col}}(c_j) \qquad \in \mathbb{R}^{n \times d} \qquad (1)$$

$$e_j = W \odot c_j + B \qquad \in \mathbb{R}^{n \times d} \qquad (2)$$

Note that each cell in a column is assigned its own weight and bias. Essentially, feature embedding can be framed as a *set-input* problem, where $\text{TF}_{\text{col}}$ serves as a hypernetwork taking as input a permutation-invariant set of cell values and generating distribution-aware weights and biases $W$ and $B$.

The operations within $\text{TF}_{\text{col}}$ unfold as follows:

$$U = \text{Lin}(c) \qquad \in \mathbb{R}^{n \times d} \qquad (3)$$

$$\text{ISAB} \begin{cases} M = \text{MAB}_1(V_I, U_{\text{train}}, U_{\text{train}}) & \in \mathbb{R}^{k \times d} \qquad (4) \\ V = \text{MAB}_2(U, M, M) & \in \mathbb{R}^{n \times d} \qquad (5) \end{cases}$$

$$W, B = \text{Lin}(V) \qquad \in \mathbb{R}^{n \times d} \qquad (6)$$

where Lin represents a linear layer, MAB denotes a multi-head attention block, and IASB stands for induced self-attention block (Figure 2a), introduced by (Lee et al., 2019).

A column $c$ (omitting the subscript $j$ for simplicity) is projected into a $d$-dimensional space ($d = 128$) via a linear layer, processed by ISAB consisting of $\text{MAB}_1$ and $\text{MAB}_2$, and passed through linear layers to generate $W$ and $B$.

ISAB reduces self-attention complexity to $\mathcal{O}(n)$ while preserving the ability to capture global information through a two-stage attention mechanism. In $\text{MAB}_1$, inducing vectors $V_I$ act as queries and attend to training samples $U_{\text{train}}$ to generate induced representations $M$. In $\text{MAB}_2$, inputs $U$ (including both training and test samples) serve as queries and attend back to $M$, enabling global information to propagate across all training samples. We use $k = 128$ inducing vectors, 4 attention heads in the MABs, and 3 ISAB blocks (only one ISAB block is shown in Equations (4) and (5) for clarity). Crucially, only train samples serve as keys and values in $\text{MAB}_1$. This ensures that the outputs of ISAB depend solely on training data, thereby preventing data leakage.

To get insights into the information captured in the learned feature embeddings, we visualize $M \in \mathbb{R}^{k \times d}$ (Equation (4)) the output of the final ISAB block of $\text{TF}_{\text{col}}$. We summarize $M$ by summing along the first dimension (i.e., aggregating the induced representations of all inducing vectors) to obtain a single vector per column, and then apply Principal Component Analysis. Figure 3 reveals that columns with similar skewness (resp. kurtosis) tend to cluster together, i.e., non-symmetric distributions are differentiated from symmetrical ones (Figure 3 left), and heavy-tailed distributions are differentiated from light-tailed ones. This suggests that $\text{TF}_{\text{col}}$ encodes distributional properties in a structured manner, and cells in a column are probably embedded to reflect their statistical role (e.g., min, max, mean, mode). Therefore, the learned feature embeddings should distinguish features based on their unique distributional properties, effectively serving as feature identifiers. This contrasts with methods that rely on semantic column names (Kim et al., 2024) or learn feature identifier vectors (Kossen et al., 2021).

### 3.3. Context-aware Row-wise Interaction

After obtaining all feature embeddings $E = [e_1, \cdots, e_m] \in \mathbb{R}^{n \times m \times d}$, a 3-layer transformer with 8 attention heads, denoted by $\text{TF}_{\text{row}}$, processes $E$ for inter-feature interactions. To aggregate the embeddings into a single vector, four learnable [CLS] tokens are prepended to each row of $E$, and their final outputs are concatenated together. We use four tokens to provide richer representations with a total embedding size of $4 \times d = 512$ for subsequent ICL, while maintaining a lower embedding size ($d = 128$) for $\text{TF}_{\text{col}}$ and $\text{TF}_{\text{row}}$ to reduce memory consumption.

We experimentally observed that $\text{TF}_{\text{row}}$ can suffer from a representation collapse issue. As mentioned earlier, features are identified by their distributional properties after column-wise embedding. Consequently, features originating from

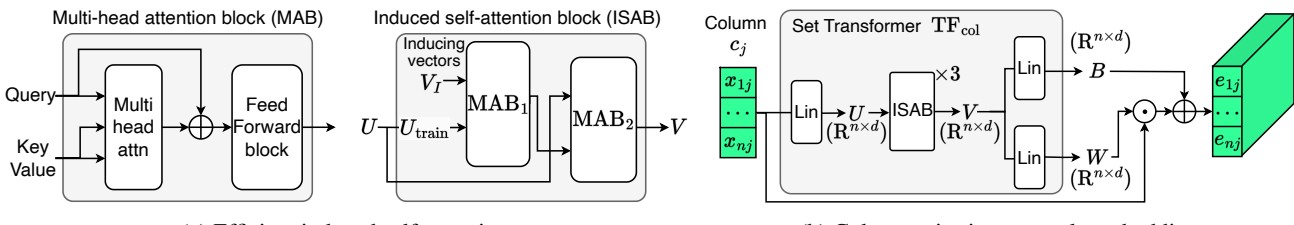

(a) Efficient induced self-attention       (b) Column-wise inter-sample embedding

*Figure 2.* **The distribution-aware column-wise embedding block in TabICL.** The overall structure (right) relies on the induced self-attention block (middle) that employs two multi-head attention blocks (left) to process incoming embeddings.

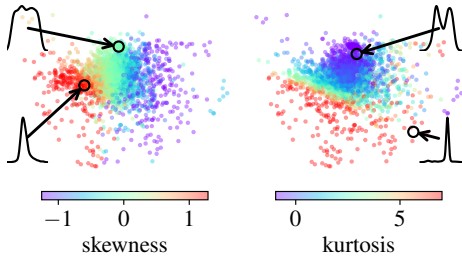

*Figure 3.* **Learned column-wise embeddings encode statistical properties of feature distributions.** Visualization of these embeddings projected onto their first two principal components for 40,000 features from synthetic datasets.

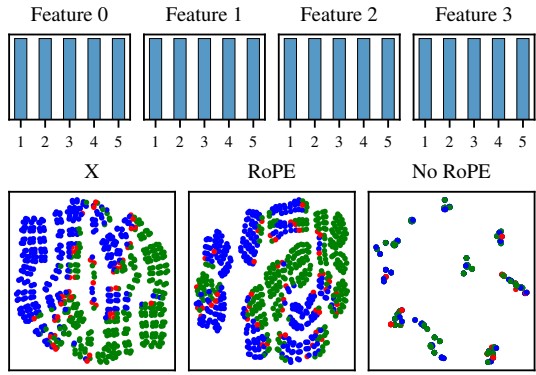

*Figure 4.* **An example of learning collapse without RoPE for the** `balance scale` **dataset.** (*Upper*) The histograms show that the four features follow the same discrete distribution. (*Lower*) The t-SNE visualization of the input features and learned row-wise embeddings $H$ with and without RoPE demonstrates that RoPE can effectively alleviate representation collapse. The colors represent 3 classes.

similar distributions thus become less distinguishable. In the extreme case where all features are drawn from the same distribution, $\text{TF}_{\text{row}}$ using the vanilla permutation-invariant self-attention cannot differentiate a sample from any of its column-permuted versions, leading to nearly identical representations for originally distinct samples, i.e., representation collapse. This phenomenon is exemplified by the `balance scale` dataset (Siegler, 1976), as shown in Figure 4. In this dataset, all features follow the same discrete distribution with only 5 possible values, making collapse highly probable. After processing with $\text{TF}_{\text{row}}$, we observe that many samples collapse to the same representation (rightmost plot), despite being originally distinct (leftmost plot).

A similar issue of representation collapse was reported in TabPFNv2. To mitigate this, Hollmann et al. (2025) introduced random feature identifier vectors for each feature and encoded groups of features collectively rather than individually. In contrast, we employ a different strategy by incorporating rotary positional embedding (RoPE, Su et al., 2023) into $\text{TF}_{\text{row}}$ to break the symmetry between identically distributed features. RoPE is widely adopted in recent LLMs (Dubey et al., 2024) and directly encodes relative positional information into the attention mechanism by rotating the query and key vectors. The rotation angle is determined by the position $p$ in the sequence and the dimension index $i$, defined as $\theta_i = p/(\text{base}^{2i/d})$, where $d$ is the embedding dimension and base is the frequency scaling factor. More

details can be found in Appendix D.

While the use of RoPE breaks permutation invariance with respect to column order, it can effectively mitigate the representation collapse. For instance, on the balance scale dataset, RoPE preserves distinct representations across different samples (Figure 4 middle). Following Xiong et al. (2023), we use a large scaling factor of 100,000 for RoPE to enhance generalization to a greater number of features than seen during training (up to 100). To approximately restore permutation invariance, TabICL adopts the same strategy as other TabPFN-like models by ensembling predictions over multiple column permutations.

### 3.4. Dataset-wise In-Context Learning

After converting all samples into embeddings $H \in \mathbb{R}^{n \times 4d}$, training labels are mapped to the same space as $H$ using one-hot encoding. The embeddings of $X$ and $y$ for the training set are added to create the final training embeddings $H_{\text{train}}$. We then process $H_{\text{train}}$ and $H_{\text{test}}$ using a 12-layer Transformer with 4 attention heads, denoted by $\text{TF}_{\text{icl}}$. The

embeddings in $H_{\text{train}}$ can attend to one another while those in $H_{\text{test}}$ can only attend to $H_{\text{train}}$. Finally, a two-layer MLP converts the outputs of $H_{\text{test}}$ into class probabilities for the test samples.

### 3.5. Computational Complexity Analysis

To characterize the computational efficiency and scalability of TabICL, we analyze the time complexity of its three components for a table with $n$ rows and $m$ features:

1. $\text{TF}_{\text{col}}$ processes each column independently using ISAB, resulting in complexity $\mathcal{O}(nkm)$, where $k$ is the fixed number of inducing vectors.
2. $\text{TF}_{\text{row}}$ applies self-attention over features and 4 [CLS] tokens per row, yielding complexity $\mathcal{O}(m^2 n)$. $\text{TF}_{\text{row}}$ is efficient since $m \ll n$ in most tabular settings.
3. $\text{TF}_{\text{icl}}$ operates on the compressed row embeddings via self-attention with complexity $\mathcal{O}(n^2)$. While quadratic in $n$, its cost is incurred once since the feature dimension is collapsed, ensuring practical efficiency.

The overall complexity of TabICL is $\mathcal{O}(m^2 n + n^2)$. In contrast, TabPFNv2 alternates column-wise and row-wise attentions without collapsing dimensions, resulting in complexity $\mathcal{O}(m^2 n + n^2 m)$. As a result, TabPFNv2 can become computationally expensive for large $n$ and moderate $m$.

## 4. Pretraining and Inference

### 4.1. Improved Pretraining Synthetic Datasets

TabICL is pre-trained exclusively on synthetic datasets. To ensure realistic dependencies among variables, we generate these datasets using structural causal models (SCMs), following the approach proposed in TabPFN (v1). We first sample a directed acyclic graph (DAG) to define dependencies, following the structure of a fully connected MLP, where each neuron corresponds to a variable. Each feature $c$ is then modeled as a function $f$ of its parent variables $\text{Pa}(c)$ in the graph, with added independent noise $\epsilon$, *i.e.*, $c = f(\text{Pa}(c)) + \epsilon$. Compared to previous work, we enrich the dataset generation in two ways: (i) we introduce tree-based SCMs to benefit from their inductive biases, and (ii) increase the diversity of modeling functions $f$.

**Tree-based generation**    Inspired by Grinsztajn (2024), we introduce tree-based SCMs to leverage their ability to model complex interactions and hierarchical dependencies between variables. We define $f$ using an XGBoost regression model, as it is widely favored by practitioners and supports multi-output regression. At each layer of the graph, an XGBoost model is trained on fake targets drawn from Gaussian noise, taking the values of the parent variables as input. The obtained predictions then become the values of the child vari-

ables. To balance data generation, we combine SCMs (70%) with tree-based SCMs (30%). Appendix C gives details and examples of generated data.

**Diversifying activation functions**    In TabPFN (v1), $f$ is defined as a random affine mapping (a linear layer) with an activation function chosen from [Identity, Tanh, Leaky ReLU, ELU]. We enrich this set with 15 additional activation functions to enhance the diversity of non-linear dependencies, introducing for example non-monotone or discontinuous functions. We also include random activation functions sampled from Gaussian processes with random kernels. Gaussian Process functions have been used for synthetic data generation for time series foundation models (Ansari et al., 2024), but not as activation functions and with different types of kernels. Finally, we added an option to use different activation functions across layers and applied standardization followed by random rescaling before each activation function. Figure C.1 visualizes the employed activation functions.

### 4.2. Curriculum Learning for Large-scale Pretraining

Similar to pretraining LLMs on shorter sentences before moving to longer ones, we gradually increase the size of synthetic datasets (i.e., the number of samples) while adjusting the micro batch size $N_{\mathcal{B}}$ used for gradient accumulation to accommodate memory constraints. We employed a three-stage procedure:

1. $N_{\mathcal{B}} = 4$ with a fixed size of 1,024 for 160K steps;
2. $N_{\mathcal{B}} = 1$ with the size randomly drawn from a log-uniform distribution between 1K and 40K over 2K steps. Activation checkpointing is enabled for datasets exceeding 10K samples, and we accordingly reduce the number of features to avoid out-of-memory issues;
3. $N_{\mathcal{B}} = 1$ with the size uniformly sampled between 40K and 60K for 50 steps, training only $\text{TF}_{\text{icl}}$ while freezing all other components.

Each step consists of 512 datasets with the number of features ($\leq 100$) and classes ($\leq 10$) randomly sampled. FlashAttention and automatic mixed precision are applied globally. The pretraining took 20 days on three A100 GPUs with 40GB memory using PyTorch (16, 3, and 1 days for stage 1, 2, and 3, respectively). Appendix E.1 gives more pretraining details.

### 4.3. Hierarchical Class-extension Strategy

We tackle many-class classification problems ($> 10$ classes) through hierarchical classification (Silla & Freitas, 2011). Specifically, we recursively and evenly partition classes into subgroups of up to 10 classes, forming a multi-level classification tree. A classification problem with $k$ classes requires

a hierarchy with depth $r = \lceil \log_{10} k \rceil$. Each internal node in the tree predicts the probability of its child groups, and each leaf predicts the probability of the classes it contains. During inference, the final probability for a given class is obtained by multiplying the probabilities across all nodes situated on the unique root-to-leaf path leading to this class. See Appendix E.3 for more details.

As noted earlier, labels are used exclusively in the final ICL block. Consequently, the hierarchical tree is constructed during dataset-wise ICL, with all sub-tasks sharing the same learned row embeddings $H$ and the same $\text{TF}_{\text{icl}}$ for ICL predictions. This kind of sharing greatly enhances the efficiency of TabICL in hierarchical classification scenarios.

### 4.4. Memory-efficient Inference

Using FlashAttention, which offers linear memory complexity with respect to sequence length, we observed that the peak activation memory can be effectively modeled by a polynomial regression during inference: $\alpha_1 \times \text{batch\_size} + \alpha_2 \times \text{seq\_len} + \alpha_3 \times \text{batch\_size} \times \text{seq\_len} + \alpha_4$. This enabled us to dynamically adjust the batch size based on sequence length and available GPU memory. The batch dimension serves different roles depending on the context: it represents the number of columns for column-wise embedding, the sample size for row-wise interaction, and the number of datasets for dataset-wise ICL. Inspired by Rajbhandari et al. (2021), intermediate activations can be offloaded to CPU and disk as needed to further reduce memory consumption. These optimizations enable TabICL to handle datasets with 100K samples and 500 features using only 5 GB of GPU memory and 32 GB of RAM. This suffices for many real-world applications. See Appendix E.2 for details.

## 5. Experiments

### 5.1. Benchmark

We evaluate TabICL on the TALENT benchmark introduced by Ye et al. (2025). It comprises 200 classification datasets (120 binary and 80 multiclass datasets) and comes with results for over 30 baseline methods. To ensure a fair comparison, we exclude 15 datasets that were used by TabPFNv2 for hyperparameter tuning and model selection (see Appendix F for more details). In what follows, we mainly focus on the 171 datasets with at most 10 classes, since these can be handled natively by TabICL and TabPFNv2.

Datasets are split into 64% training, 16% validation, and 20% test data. While most models rely on the validation set for early stopping and hyperparameter tuning, TabICL and TabPFNv2 do not. Nonetheless, we opted to train TabICL and TabPFNv2 using the training data only, which places them at a disadvantage. On the flip side, both TabICL and TabPFNv2 leverage ensembles, unlike the other deep learn-

ing models. In Appendix H, we show that increasing ensemble size improves performance for both TabPFNv2 and TabICL, with TabPFNv2 benefiting more from ensembling. A fair comparison would require training the other models as an ensemble as well. However, this is computationally expensive, was not part of the original benchmark, and is not implemented in these experiments.

Both TabICL and TabPFNv2 average over 32 predictions obtained by randomly shuffling columns and classes and using different pre-processors. TabICL applies $z$-normalization with or without power transformation to all input features. See Hollmann et al. (2025) for details on TabPFNv2. To avoid out-of-memory issues, we subsample training sets to 30K samples for TabPFNv2, while TabICL could predict on all datasets without subsampling. In addition, we disable automatic mixed precision across all methods during inference to ensure reproducibility and consistent time measurements.

### 5.2. Results

**TabICL obtains state-of-the-art accuracy.** Figure 5 shows accuracies relative to the tuned MLP for all models. TabICL obtains the best median relative accuracy across all datasets while being much faster than traditional state-of-the-art models: the geometric mean training+inference time per 1K samples for TabICL is 1.1 seconds, while tuning CatBoost on a CPU takes around 3 minutes, and RealMLP and ModernNCA on a GPU take around 7 minutes. Looking at the ranks of each method based on the obtained accuracies, the critical difference diagram in Figure I.1 shows that TabICL and TabPFNv2 outperform competitors by a wide margin, while the difference between the two of them is not statistically significant.

**Speedup over TabPFNv2.** Figure 6 shows that TabICL is 1.5× faster than TabPFNv2 on small datasets and 3-10× faster on large datasets. This is facilitated by the hybrid architecture of TabICL, using fewer of the expensive row-wise and column-wise attention layers with smaller embedding dimension before ICL on tokenized rows. On a dataset with 10,000 samples and 100 features, TabICL takes roughly 20 seconds versus 1 minute 40 seconds for TabPFNv2, while for a dataset of 1000 samples and 10 features, TabICL takes 1 second compared to 2 seconds for TabPFNv2. The empirical speedup of TabICL over TabPFNv2 also corroborates the complexity analysis in Section 3.5. In Figure B.1, we fit simple scaling laws to predict the runtime of TabICL and TabPFNv2 based on the number of samples and features. These scaling laws show that the average speedup of TabICL remains around five-fold for large datasets.

**TabICL enables ICL for large datasets.** While TabPFNv2 achieves excellent performance on datasets up to 10K samples, it has only been pre-trained with up to 2048

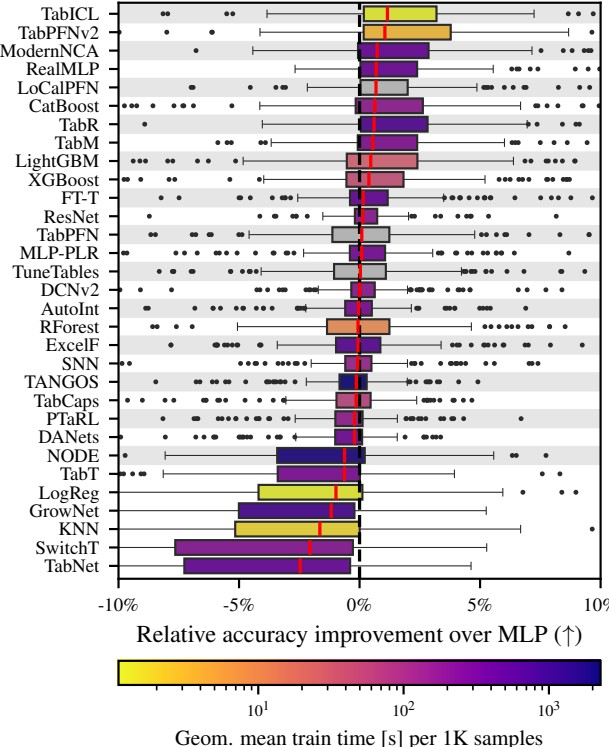

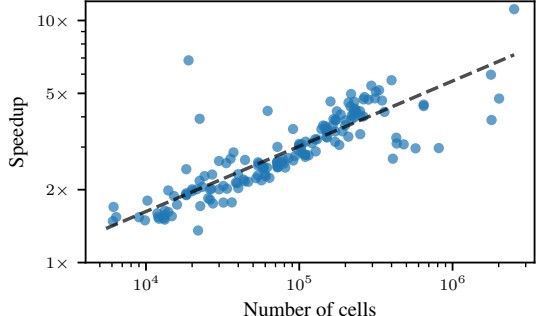

Figure 6. **Speedup of TabICL vs. TabPFNv2 on datasets with less than 30K samples and at most 10 classes.**

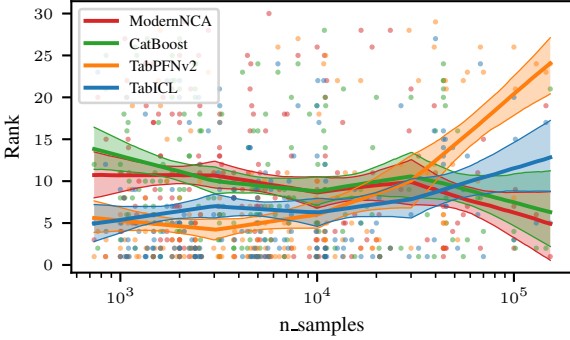

Figure 7. **Model rankings as a function of sample size.** Each point is the rank of one method on one dataset. Lower rank is better. The lines show the bootstrap median and 10% / 90% bootstrap confidence intervals of a piecewise linear fit.

Figure 5. **Accuracy and training/inference times on the TAL-ENT benchmark up to 10 classes.** The red bar indicates the median relative accuracy improvements over MLP across all datasets for each method. For TabICL and TabPFNv2, the time reported corresponds to training+inference on an A100 GPU. For the other models, it includes both training and hyperparameter tuning. Since Ye et al. (2025) track only the training time using the best hyperparameters found after 100 tuning steps, we approximate the total training time by multiplying this value by 100. For TabPFN (v1), LoCalPFN, and TuneTables, we do not report the time since the original time measurement does not include the inference time.

training samples and can fail on datasets above 30K samples due to its memory usage. Figure 7 shows that unlike TabPFNv2, the performance of TabICL remains strong for larger datasets. While ICL is often used in few-shot settings, this demonstrates that foundation models can compete with the best deep learning or tree-based models, even in large-sample regimes where the latter have the most potential. Additional results in Appendix B demonstrate that TabICL also performs well for many classes, features, and high ratios of categorical features. To enhance the performance of TabPFNv2 on large datasets, Hollmann et al. (2025) proposed a random forest extension: during training, the data is partitioned using decision trees, and TabPFNv2 is fitted at each leaf to enable localized predictions at inference time. We found that this extension can significantly improve the performance of both TabPFNv2 and TabICL on large datasets (refer to Appendix G for more details).

**TabICL produces reliable probabilities.** In many practical situations, the quality of the estimated probabilities is crucial for decision-making. Therefore, we also report the log loss (a.k.a. cross-entropy loss), which is a proper scoring rule and therefore rewards accurate prediction of probabilities (Gneiting & Raftery, 2007). Since TabICL does not leverage hyperparameter tuning, its predictions are not specifically optimized towards accuracy. The critical difference diagrams in Appendix I show that TabICL and TabPFNv2 significantly outperform accuracy-tuned competitors on the log loss, though the difference between the two is not significant, indicating that both produce more reliable probability estimates than the other models.

**TabICL remains effective with more than 10 classes.** On datasets with more than 10 classes, we apply the hierarchical classification strategy from Section 4.3 to TabICL. Thanks to its use of label-independent row embeddings that can be shared between all sub-classifiers, TabICL scales efficiently to a large number of classes. Figure 8 shows that TabICL still achieves the second best result on these datasets in terms of mean normalized accuracy, while TabPFNv2 cannot natively handle more than 10 classes.

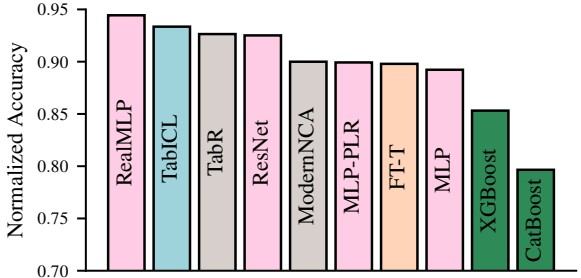

Figure 8. **Normalized accuracy across the 12 datasets with more than 10 classes.** Colors identify tree-based models (green), deep learning models (pink), retrieval models (gray), transformers (orange), and in-context models (blue).

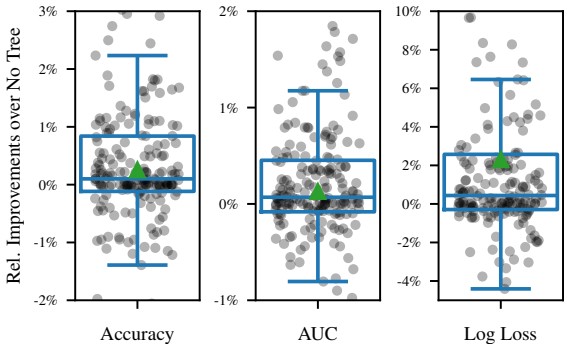

Figure 9. **Relative improvements of TabICL pretrained on prior datasets with versus without tree-based SCMs across 200 datasets.** Green triangles denote the mean relative improvements. To mitigate computational overhead, we pretrained TabICL for 20K steps following the stage 1 setup of curriculum learning.

### 5.3. Ablation Studies

**Tree-based SCMs prove beneficial.** Incorporating the inductive biases of tree-based models during pretraining improve the performance of TabICL, as shown in Figure 9.

**Curriculum learning effectively improves the performance of TabICL on large datasets.** Across the three stages of curriculum learning, the average rank of TabICL improves from 11.4 (ninth place), to 7.46 (second), and finally to 6.95 (first), as shown in Figure E.2. These results show the effectiveness of curriculum learning. However, we also observe a slight performance drop on some small datasets during curriculum learning, as shown in Figure 10.

## 6. Conclusion

We introduce TabICL, a novel tabular foundation model that extends the scalability of existing tabular foundation models by an order of magnitude. Evaluated on datasets with up to 100K training samples, it delivers excellent performance without hyperparameter tuning, making it nearly

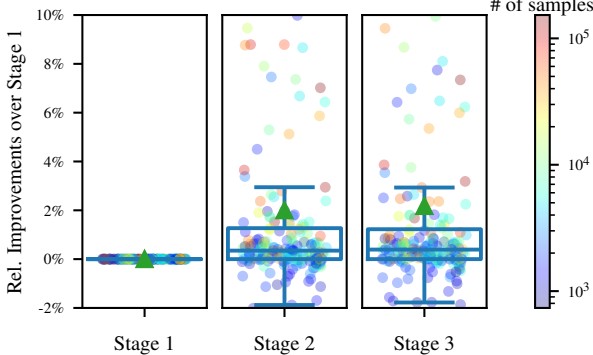

Figure 10. **Relative accuracy improvements over Stage 1 across 200 classification datasets.** Green triangles denote the mean relative improvements.

two orders of magnitude faster than other tabular methods requiring hyper-parameter tuning. TabICL achieves this through a hybrid architecture and memory-saving optimizations. Compared to the newly released and leading tabular foundation model TabPFNv2, TabICL achieves comparable performance while being more scalable and faster.

**Limitations** (1) Like other foundation models, TabICL suffers from slow inference speed, though TabPFNv2 has shown caching can alleviate this. Currently, TabICL is limited to classification, but regression can be treated with similar methodology, as Hollmann et al. (2025) showed. (2) A defining characteristic of tabular data is that columns do not have a natural ordering. Ideally, tabular methods should be invariant to column permutations. However, TabICL violates this invariance due to the use of RoPE. (3) Our evaluation inherits the strengths and weaknesses of the TALENT benchmark. Notably, TALENT trains single models tuned using the holdout method, while ensembling models evaluated using cross-validation can improve performance. Bagging increases computational cost, whereas TabICL can simply be trained on the full data without needing a validation set. (4) TALENT employs mean imputation for handling missing values. However, it might be better to allow TabPFNv2 to handle missing values internally despite the low proportion of missing values in TALENT. (5) We did not provide categorical information to TabPFNv2. Though TabPFNv2 infers categorical features internally, providing this information explicitly could enhance its performance.

**Outlook** In-context learning for tabular data was originally introduced as a speedup for small tables (Hollmann et al., 2022). As the expressive power of a forward pass in a transformer may seem limited, one might wonder if in-context learning loses its edge given sufficient data. We find that even with large data, pre-training induces implicit priors that give in-context transformers a competitive advantage.

# Acknowledgements

We thank Lennart Purucker and Samuel Müller for interesting discussions. We are also grateful to Si-Yang Liu and Han-Jia Ye, the authors of the TALENT benchmark, for providing information and clarifying our questions. Finally, we thank the anonymous ICML reviewers for their valuable feedback and suggestions.

# Impact Statement

This paper presents work whose goal is to advance the field of Machine Learning. There are many potential societal consequences of our work, none which we feel must be specifically highlighted here.

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

# A. Systematic Comparison between TabICL and TabPFNv2

This appendix provides a systematic comparison between TabICL and TabPFNv2, two tabular foundation models that leverage In-Context Learning (ICL) to make predictions. Both models are pretrained on synthetic datasets and aim to deliver strong performance on unseen real-world tabular datasets.

## A.1. Architecture

**Core mechanism.** (1) TabICL employs a two-stage architecture, where the first stage uses a column-then-row attention mechanism to build fixed-dimensional embeddings and effectively collapses the column dimension to reduce computational complexity for the subsequent ICL stage. (2) TabPFNv2 utilizes a two-way attention mechanism that alternates between column-wise and row-wise attentions. These attentions operate on the input table across original dimensions without any intermediate collapse.

**Representation collapse.** When features have similar distributions, both TabICL and TabPFNv2 may suffer from the representation collapse issue. (1) To alleviate this problem, TabICL incorporates Rotary Positional Embedding (RoPE) into the transformer used in row-wise interaction. RoPE helps break the symmetry between identically distributed features by encoding relative positional information. (2) TabPFNv2 addresses this by introducing random feature identifier vectors for each feature. It also encodes groups of features collectively rather than individually.

**Label fusion.** TabICL and TabPFNv2 differ significantly in when they incorporate label information. (1) TabICL employs a late fusion strategy. TabICL processes input features independently of their labels during row-wise embedding and column-wise interaction. Labels are utilized exclusively in the final ICL stage. (2) TabPFNv2 employs an early fusion strategy for label information. TabPFNv2 directly concatenates the input features with the labels of the training samples, processing them together from the very beginning and throughout its layers.

## A.2. Pretraining

**Prior generation.** Both TabICL and TabPFNv2 are pretrained exclusively on synthetic prior datasets. (1) TabICL builds on the previous prior generation of TabPFN (v1), which relies on Structural Causal Models (SCMs). TabICL introduces two more extensions: broader nonlinear modeling capabilities of SCMs and the integration of tree-based SCMs to incorporate the inductive biases of tree-based models. (2) TabPFNv2 constructs diverse Directed Acyclic Graphs (DAGs) through a sophisticated procedure (i.e., growing networks with redirection sampling method) and incorporates various computational mappings at graph edges, including small neural networks, categorical feature discretization, and decision trees. However, the code for its prior generation is not open-source.

**Dataset size.** (1) TabICL was pretrained using curriculum learning that progressively scales the number of samples from 1,024 to 60K. The number of features was sampled uniformly up to 100, and the number of classes was sampled uniformly up to 10. (2) TabPFNv2 was pretrained on synthetic datasets containing up to 2,048 training samples (uniformly sampled). The number of features was drawn from a beta distribution and then linearly scaled to the range from 1 to 160, while the number of classes was also uniformly sampled up to 10.

**Pretraining scale.** (1) TabICL was pretrained over approximately 82 million synthetic datasets. (2) TabPFNv2 was trained on a larger corpus, comprising approximately 130 million synthetic datasets.

## A.3. Scalability and Efficiency

**Dataset size handling.** (1) At inference, TabICL is capable of handling tables with 100K samples and 500 features within 5GB of GPU memory, and can deal with any number of classes using hierarchical class-extension strategy. (2) TabPFNv2 excels on tables with up to 10K samples and 500 features, and can only deal with up to 10 classes.

**Computational complexity.** Given a table with $n$ rows and $m$ columns, the time complexity of TabICL is $\mathcal{O}(m^2 n + n^2)$, while the time complexity of TabPFNv2 is $\mathcal{O}(m^2 n + n^2 m)$. Our extensive experiments demonstrate that TabICL is systematically faster than TabPFNv2.

# B. Further Experiments

**Predicting the runtime of TabICL and TabPFNv2.**   To obtain rough predictions of the time for a forward pass (training and inference) in TabICL and TabPFNv2, we leverage the runtime complexity of the column- and row-wise attention modules. As discussed in Section 3.5, for a table with $n$ rows and $m$ columns, the runtime complexity is $O(n^2m + nm^2) = O(nm(n + m))$ for TabPFNv2, while it is $O(n^2 + nm^2)$ for TabICL. For simplicity, we use the term $nm(n + m)$ from the complexity of TabPFNv2, plot this quantity on the $x$-axis of Figure B.1, and fit models of the form

$$\text{time} = \alpha + \beta(nm(n + m))^\gamma$$

with MSLE loss, that is, a MSE loss between the log-transformed time and the log-transformed prediction. To facilitate a simpler comparison, we fix $\gamma := 0.8$ for both models since it yields a good fit. We suspect that $\gamma = 1$ should be used asymptotically for TabPFNv2, but then more additional terms would be needed to obtain a good fit. In this model, the speedup of TabICL over TabPFNv2 for large datasets approaches 5, while it is 1.4 for small datasets.

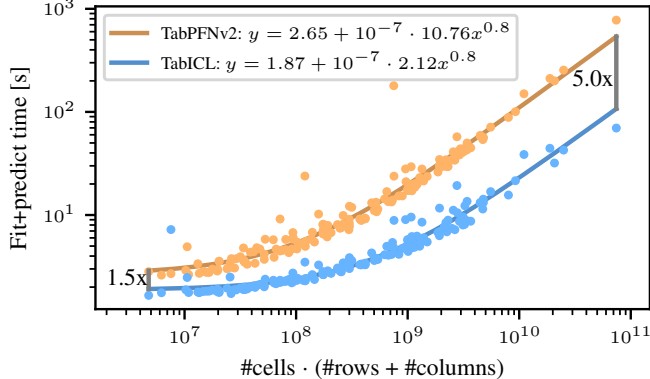

*Figure B.1.* **Training+inference times of TabICL and TabPFNv2.** Each point represents the time on one dataset on an A100 GPU. We only use datasets with less than 30,000 samples because TabPFNv2 is applied with subsampling on larger datasets to avoid RAM overflow.

**Metafeatures.**   We analyze the dependence of the ranks of TabICL, TabPFNv2, CatBoost, and ModernNCA based on different dataset metafeatures. To this end, we fit piecewise linear regression models with pre-defined nodes to predict the ranks of these models depending on a single metafeature.

Figure B.2 shows the scaling with the number of classes. The performance of TabICL deteriorates on datasets with three classes but not on datasets with more classes. It is unclear if this is really due to the number of classes or caused by a different characteristic present in the three-class datasets on the benchmark.

Figure B.3 shows the scaling with the number of features. These plots show that TabICL behaves well for large numbers of features even though it tokenizes entire rows in the middle of the network before seeing any labels.

Figure B.4 shows the dependence on the ratio of categorical to numerical variables. In general, the performance of TabICL and TabPFNv2 deteriorates somewhat in the presence of many categorical variables. However, it is notable that TabICL performs slightly better than TabPFNv2 on such datasets even though TabPFNv2 has a more sophisticated categorical feature generation in its prior.

# C. Synthetic Datasets for Pretraining

## C.1. SCM prior with more activation functions

In the TabPFN prior, we replace the activation layers by the following sequence of layers:

- A standardization layer that standardizes each feature across the batch (samples) dimension

- A random rescaling layer that, for each neuron $i$, computes $x_i \leftarrow \exp(2a)(x_i + b)$, with $a, b \sim \mathcal{N}(0, 1)$ sampled once per layer.

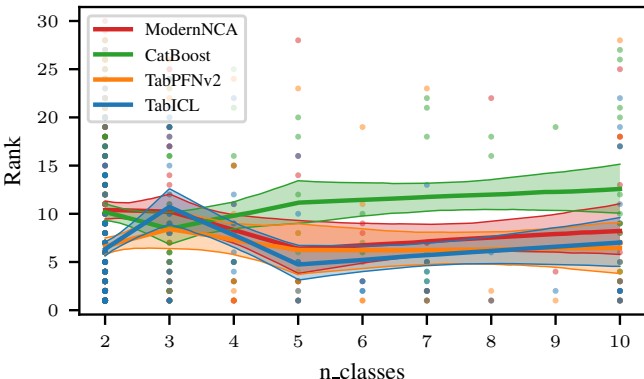

*Figure B.2.* **Dependency of the benchmark rank on the number of classes.** Each point is the rank of one method on one dataset. Lower rank is better. The lines show the bootstrap median and 10% / 90% bootstrap confidence intervals of a piecewise linear fit.

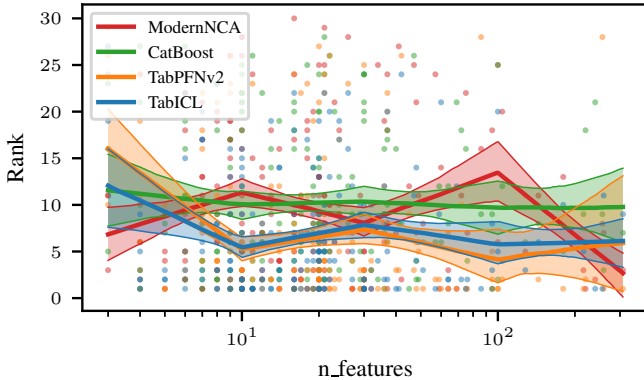

*Figure B.3.* **Dependency of the benchmark rank on the number of features.** Each point is the rank of one method on one dataset. Lower rank is better. The lines show the bootstrap median and 10% / 90% bootstrap confidence intervals of a piecewise linear fit.

- A random activation function. With probability $1/2$, each layer uses the same type of activation function, otherwise each layer samples the type of activation function independently.

On top of the original activation functions {Identity, tanh, LeakyReLU, ELU}, we add the following activation functions:

- ReLU

- ReLU6 (Krizhevsky)

- SELU (Klambauer et al., 2017)

- SiLU (Hendrycks & Gimpel, 2016)

- Softplus

- $\mathrm{Hardtanh}(x) = \max(-1, \min(1, x))$

- Signum function

- Sine

- $\mathrm{RBF}(x) = \exp(-x^2)$

- Exponential function

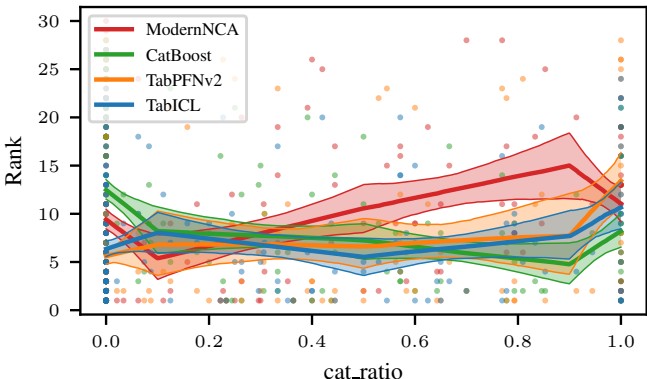

*Figure B.4.* **Dependency of the benchmark rank on the ratio of categorical to overall features.** Each point is the rank of one method on one dataset. Lower rank is better. The lines show the bootstrap median and 10% / 90% bootstrap confidence intervals of a piecewise linear fit.

- $f(x) = \sqrt{|x|}$

- $f(x) = 1_{|x| \leq 1}$

- $f(x) = x^2$

- $f(x) = |x|$

- A random function $f(x) = \phi(x)^\top z$, where $z \sim \mathcal{N}(0,1)$ and the feature map $\phi$ is defined randomly as

$$\phi(x) := \frac{\mathbf{w}}{\|\mathbf{w}\|_2} \odot \sin(\mathbf{a}x + \mathbf{b}) \in \mathbb{R}^N,$$
$$N := 256,$$
$$b_i \sim \mathcal{U}[0, 2\pi],$$
$$a_i \sim \mathcal{U}[0, N],$$
$$w_i := a_i^{-\exp(u)},$$
$$u \sim \mathcal{U}[0.7, 3.0].$$

Here, all random parameters are drawn once per layer, and $\odot$ is an element-wise product. This is motivated by the fact that for fixed feature map $\phi$, the random function $f(x) = \phi(x)^\top z$ is a Gaussian process with covariance kernel $k(x, x') = \phi(x)^\top \phi(x')$. The design of $\phi$ is inspired by random Fourier features (Rahimi & Recht, 2007). The randomly drawn exponent $-\exp(u)$ leads to different decays of coefficients with increasing frequency, which produces different levels of smoothness for the sampled function as shown in Figure C.1 (right) (Da Costa et al., 2023). This activation function applies standardization directly before the random function, such that the random rescaling has no effect.

Here, the random function is sampled with a probability ten times higher than the other activation functions to account for the fact that it can represent many different functions.

Figure C.1 visualizes the used activation functions. Figure C.2 shows datasets from the resulting prior.

### C.2. Tree-based SCM prior

The tree-based SCM prior replaces the linear and activation layers in the SCM prior by XGBoost models fitted on random data. More specifically, these models are generated as follows:

- Sample `n_estimators` and `max_depth` independently as $\min\{4, 1 + \text{Exponential}(\lambda = 0.5)\}$ and $\min\{4, 2 + \text{Exponential}(\lambda = 0.5)\}$, respectively.

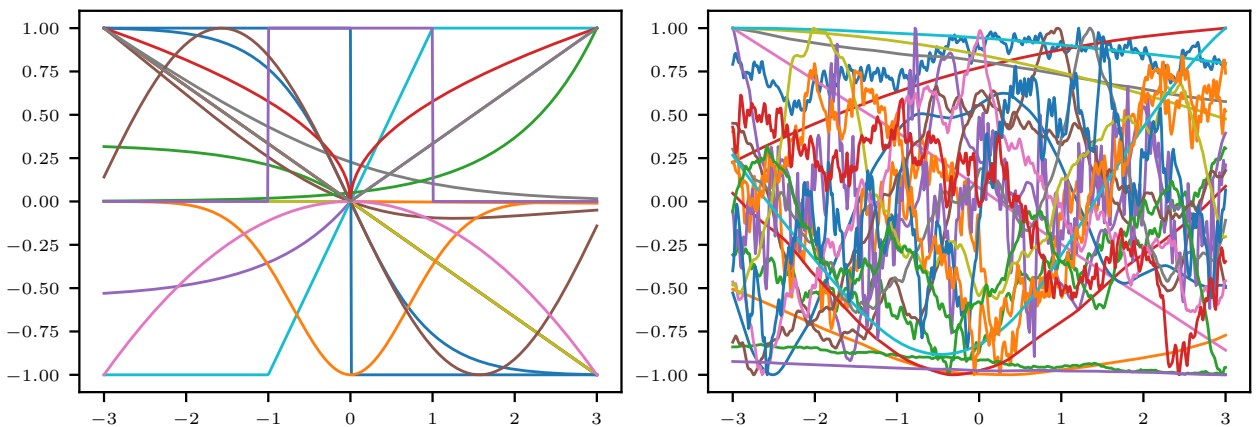

*Figure C.1.* **Activation functions from the SCM prior.** Left: Non-random activation functions from the SCM prior without standardization and random rescaling. Activation functions are randomly flipped along the $x$- and/or $y$-axis to fully utilize the space in the plot. Right: Random instantiations of the random activation function from the SCM prior, including the automatic standardization of the inputs.

*Figure C.2.* **Randomly generated 2D datasets from the SCM prior.** The color corresponds to the class label.

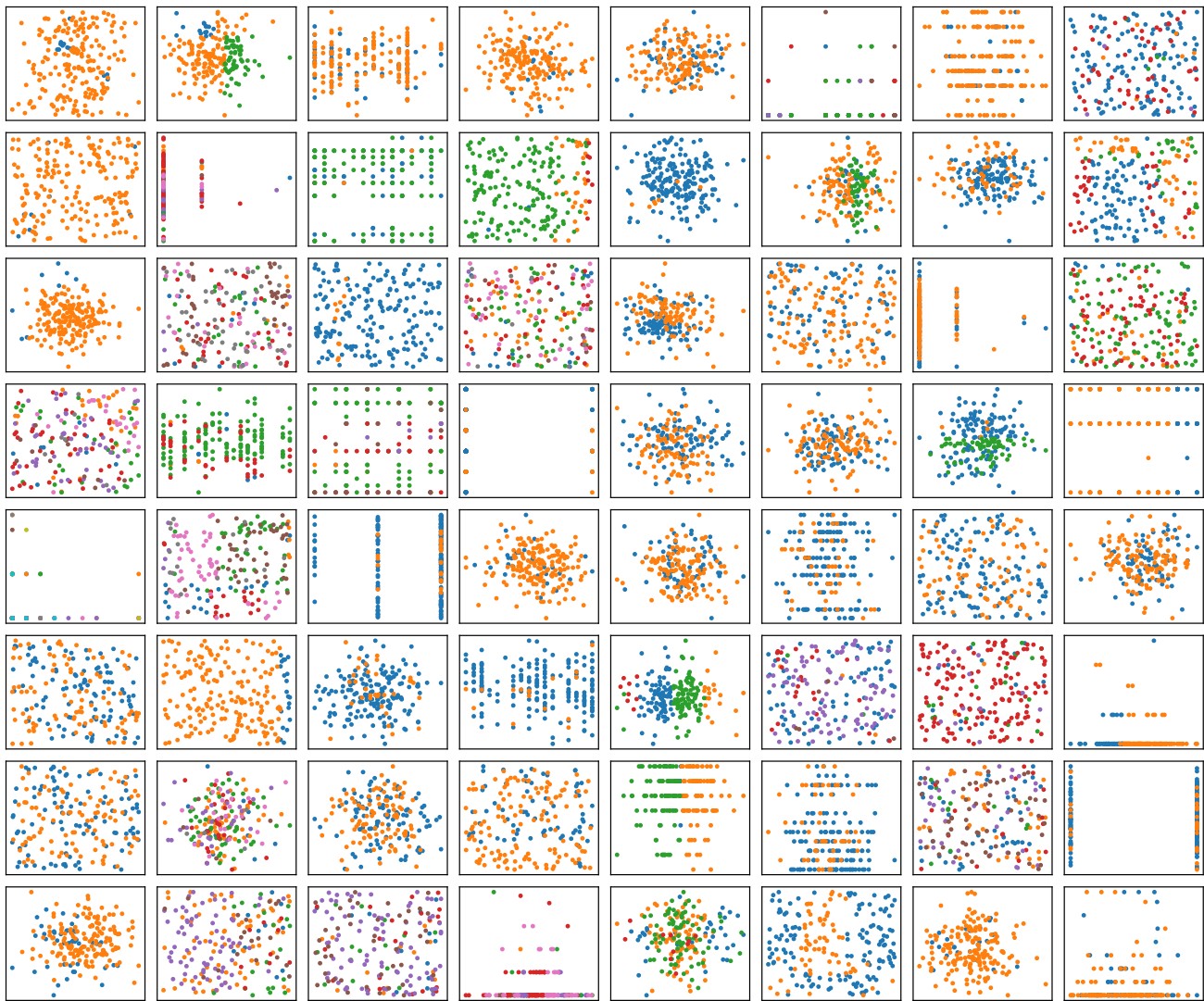

*Figure C.3.* **Randomly generated 2D datasets from the tree-based SCM prior.** The color corresponds to the class label.

- For each layer with $n$ input and $m$ output neurons, fit an XGBoost multi-output regressor with the parameters above on the layer inputs $x_i$ with standard normal targets $y_i \in \mathbb{R}^m$, then use the fitted model to predict on the given inputs.

Figure C.3 shows datasets from the tree SCM prior.

## D. Rotary Positional Embedding

Rotary Positional Encoding (RoPE) is a technique used in Transformer models to represent positional information in the input data. RoPE encodes positional information directly into the attention mechanism through a rotation matrix applied to the query and key vectors in self-attention. Given a position $p$, a frequency $\omega$, and a vector $x$ (query or key vector):

$$x_p = \text{RoPE}(x, p) = R(p)x$$

where $R(p)$ is a rotation matrix applied to the vector $x$, encoding positional information using sinusoidal functions. The rotation matrix $R(p)$ is based on a sinusoidal function and applies rotations independently to 2-dimensional subspaces of $x$.

For each 2D pair of vector components $(x_{2i}, x_{2i+1})$, the rotation is defined as:

$$R(p) \begin{bmatrix} x_{2i} \\ x_{2i+1} \end{bmatrix} := \begin{bmatrix} \cos(\theta_i) & -\sin(\theta_i) \\ \sin(\theta_i) & \cos(\theta_i) \end{bmatrix} \begin{bmatrix} x_{2i} \\ x_{2i+1} \end{bmatrix},$$

$$\theta_i := \frac{p}{100000^{2i/d}}$$

where $d$ is the dimensionality of the embedding and $\omega_i = 10000^{2i/d}$ determines the frequency for each dimension. The above equation indicates that for each dimension pair $(2i, 2i+1)$, the vector is rotated by an angle proportional to the position $p$ and the frequency $\omega_i$. RoPE directly modifies the query $Q$ and key $K$ vectors in the self-attention mechanism:

$$\text{Attention}(Q, K, V) = \text{softmax}\left(\frac{(R(p_Q)Q) \cdot (R(p_K)K)^T}{\sqrt{d}}\right) V \,.$$

Because relative positional information is preserved in the inner product, the attention scores naturally encode the relative distance between positions $p_Q$ and $p_K$.

Barbero et al. (2024) provides an alternative perspective on RoPE that aligns well with its use in our work. At lower indices $i$, the rotation angle changes more rapidly with increasing $p$, resulting in high-frequency oscillations that resemble random noise and encode positional information. Conversely, at higher indices $i$, the rotation angle changes more slowly with increasing $p$, producing stable values that carry semantic information. In our case, RoPE effectively introduces noise to each feature as its identifier in a controlled, predictable, and generalizable manner.

RoPE is also claimed to exhibit long-term decay, where tokens become less correlated as their relative distance increases. However, this claim is questioned by Barbero et al. (2024), as it relies on an unrealistic oversimplification that queries and keys are equal.

## E. Setup of TabICL

### E.1. Pretraining details

As outlined in the main paper, we employ curriculum learning to progressively increase the size of synthetic datasets (i.e., the number of samples) during pretraining. This process unfolds in three stages by adjusting the micro-batch size used for gradient accumulation:

1. $N_{\mathcal{B}} = 4$ with a fixed size of 1,024 the first 160K steps.
2. $N_{\mathcal{B}} = 1$ with the size randomly drawn from a log-uniform distribution between 1K and 40K over 2K steps. Activation checkpointing is enabled for datasets exceeding 10K samples, and we accordingly reduce the number of features to avoid out-of-memory issues.
3. $N_{\mathcal{B}} = 1$ with the size uniformly sampled between 40K and 60K for 50 steps, training only $\text{TF}_{\text{icl}}$ while all other components remain frozen.

Each step comprises 512 datasets. In the first stage, all datasets contain an equal number of samples. In the second and third stages, datasets in each micro-batch have the same number of samples, but the number of samples varies between different micro-batches. We use Adam (Kingma & Ba, 2014) and clip the gradient norm to 1. The learning rate schedules for pretraining are shown in Figure E.1, including:

- Cosine decay with restarts for stage 1

- Polynomial decay for stage 2 and the learning rate is given by $(lr_{\text{init}} - lr_{\text{end}})(1 - \text{step}/T)^2 + lr_{\text{end}}$, where $lr_{\text{init}} = 2e\text{-}5$, $lr_{\text{end}} = 5e\text{-}6$, and the number of steps $T$ is 2,000.

- Constant learning rate for stage 3

In addition, Figure E.2 shows critical difference diagrams across the three stages of curriculum learning. We observe that average rank of TabICL improves consistently from 11.4 (ninth place), to 7.46 (second), and finally to 6.95 (first). This provides strong empirical evidence for the effectiveness of curriculum learning.

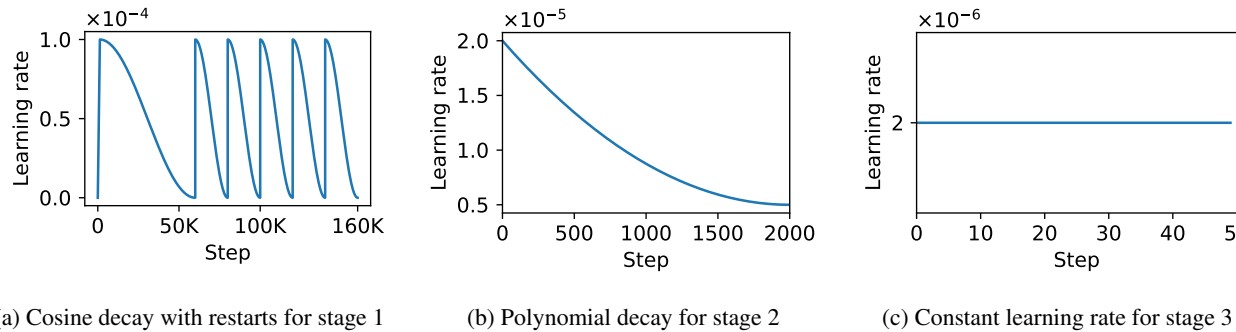

(a) Cosine decay with restarts for stage 1        (b) Polynomial decay for stage 2        (c) Constant learning rate for stage 3

*Figure E.1.* **Learning rate schedules for 3 pretraining stages.**

### E.2. Memory-efficient inference

By employing FlashAttention, we have observed that the inference peak GPU memory consumption of the three transformers of TabICL for column-wise embedding $TF_{col}$, row-wise interaction $TF_{row}$, and dataset-wise ICL $TF_{icl}$ can be well approximated through the following polynomial regression:

$$\text{MEM} = \alpha_1 \times \text{batch\_size} + \alpha_2 \times \text{seq\_len} + \alpha_3 \times \text{batch\_size} \times \text{seq\_len} + \alpha_4$$

It is important to note that the specific meanings of batch size and sequence length vary across different transformers, as outlined in Table E.1.

|  | **Batch Size** | **Sequence Length** |
|---|---|---|
| $TF_{col}$ | Number of features | Number of samples |
| $TF_{row}$ | Number of samples | Number of features |
| $TF_{icl}$ | Number of datasets | Number of samples |

*Table E.1.* Notions of batch size and sequence length for different transformers

Given an input $X \in \mathbb{R}^{b \times n \times m}$, where $b$, $n$, and $m$ represent the number of datasets, the number of samples, and the number of features, respectively, $X$ is first reshaped to $\mathbb{R}^{(b \times m) \times n}$ and processed by $TF_{col}$ to get $E = \mathbb{R}^{(b \times m) \times n \times d}$. Subsequently, $E$ is reshaped to $\mathbb{R}^{(b \times n) \times m \times d}$ and passed to $TF_{row}$, which generates $H \in \mathbb{R}^{b \times n \times 4d}$. Finally, $H$ is fed into $TF_{icl}$ to predict the test set entirely through ICL. We can see that it is necessary to set appropriate batch sizes for different transformers in order to efficiently utilize GPU resources and avoid out-of-memory errors. This is precisely where the aforementioned polynomial regression comes into play.

To this end, we first systematically tracked peak GPU memory usage of different transformers by varying both batch size and sequence length on a A100 GPU with 40GB memory, and then we fit the parameters of the above polynomial regression to the tracked data, as shown below:

$$\text{MEM}_{col} = (0.0708 \times \text{batch\_size}) + (7.29 \times 10^{-6} \times \text{seq\_len}) + (0.00391 \times \text{batch\_size} \times \text{seq\_len}) + 137.62$$

$$\text{MEM}_{row} = (-2.07 \times 10^{-5} \times \text{batch\_size}) + (2.27 \times 10^{-4} \times \text{seq\_len}) + (0.00537 \times \text{batch\_size} \times \text{seq\_len}) + 138.54$$

$$\text{MEM}_{icl} = (-0.260 \times \text{batch\_size}) + (4.77 \times 10^{-7} \times \text{seq\_len}) + (0.0195 \times \text{batch\_size} \times \text{seq\_len}) + 140.58$$

The estimated memory is measured in megabytes (MB).

In addition to adjusting the batch size, we also offload intermediate activations to the CPU or disk as needed to further alleviate GPU memory constraints. Figure E.3 illustrates the CPU and GPU memory consumption for a large dataset containing 100K samples and 500 features (80% training set and 20% test set). As shown, only 5GB of GPU memory and 25GB of CPU memory are utilized, making this a highly affordable computational setup.

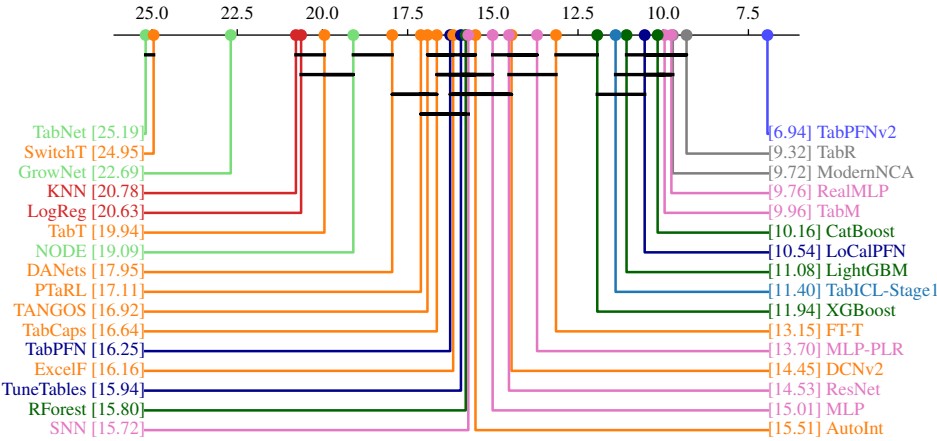

(a) Critical difference diagram (based on accuracy) after the stage 1 of curriculum learning

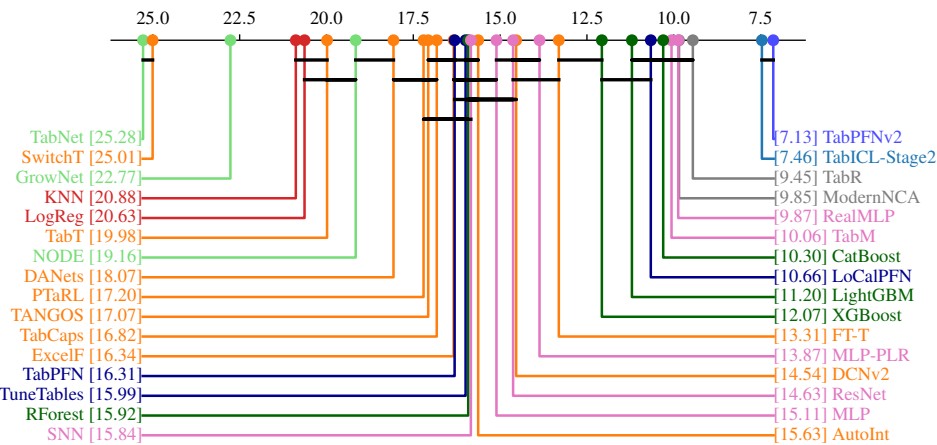

(b) Critical difference diagram (based on accuracy) after the stage 2 of curriculum learning

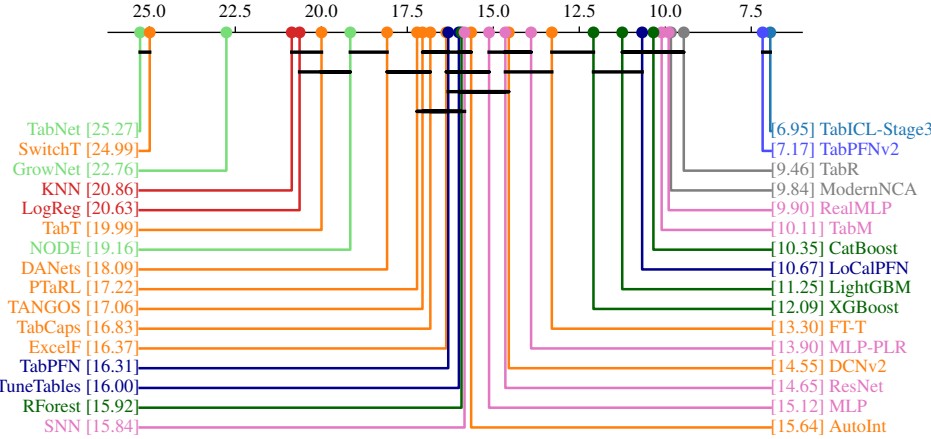

(c) Critical difference diagram (based on accuracy) after the stage 3 of curriculum learning

*Figure E.2.* **Critical difference diagrams for different stages of curriculum learning**

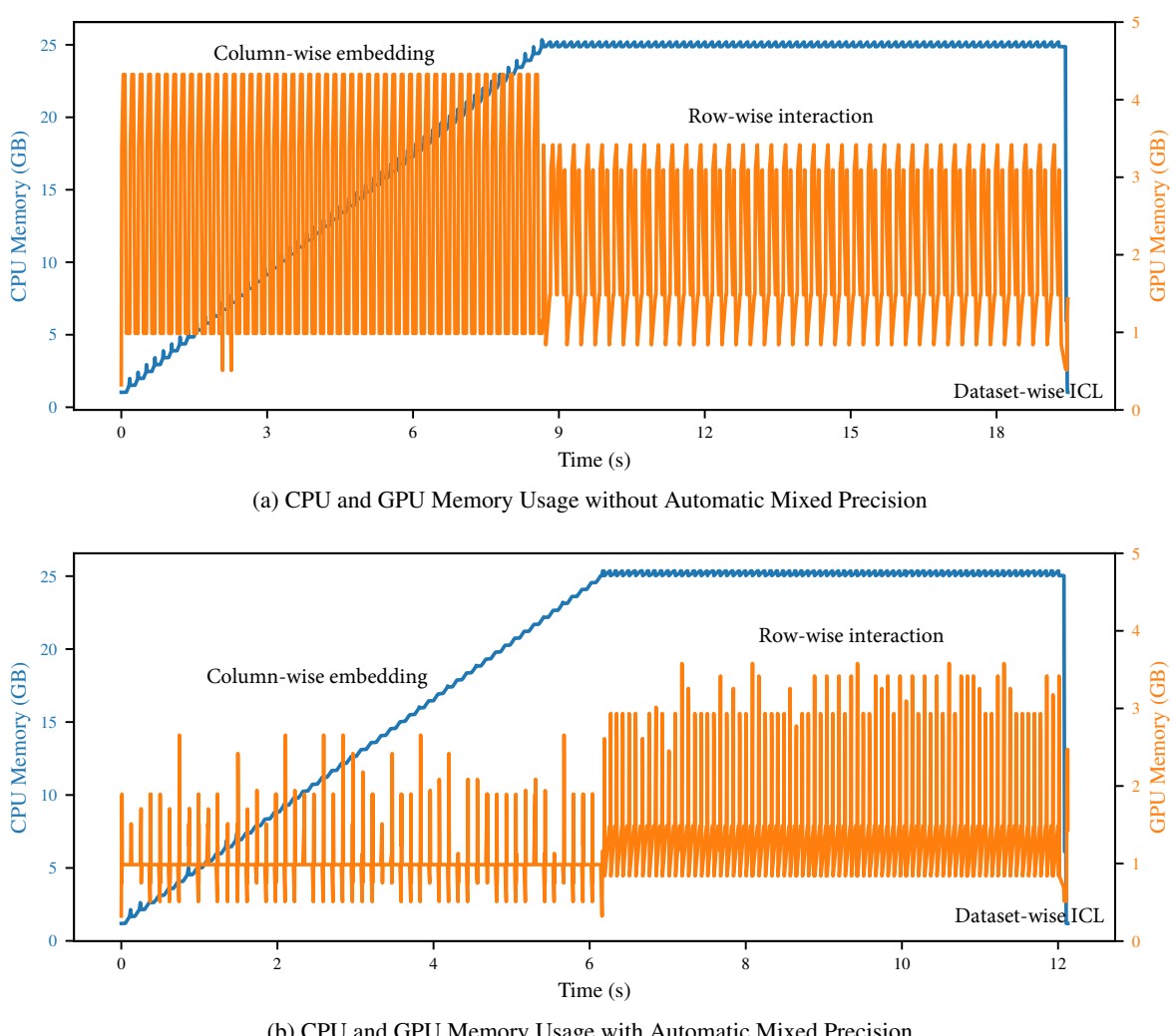

(a) CPU and GPU Memory Usage without Automatic Mixed Precision

(b) CPU and GPU Memory Usage with Automatic Mixed Precision

*Figure E.3.* **CPU and GPU memory consumption during inference for a dataset with 100K samples and 500 features.** During dataset-wise ICL, there is only a single batch (i.e., a single dataset), which corresponds to a sharp spike in GPU usage at the final stage of computation.

Figure E.4 shows the CPU and GPU memory consumption for a larger dataset containing 500K samples and 500 features (80% training set and 20% test set). As depicted, the computation requires less than 14GB of GPU memory. The CPU memory usage reaches approximately 120GB, which, however, can be significantly reduced through optional disk offloading via memory mapping.

We can observe that GPU memory consumption exhibits periodic fluctuations during the column-wise embedding and row-wise interaction phases. This is because the entire dataset is automatically divided into multiple batches, with the batch size determined dynamically based on the polynomial regression mentioned earlier. Additionally, during column-wise embedding, the output of TF$_{col}$ is progressively offloaded to CPU. As a result, we can see an incremental increase in CPU memory usage throughout this stage.

We can also see that enabling automatic mixed precision highly reduces both memory consumption and computation time.

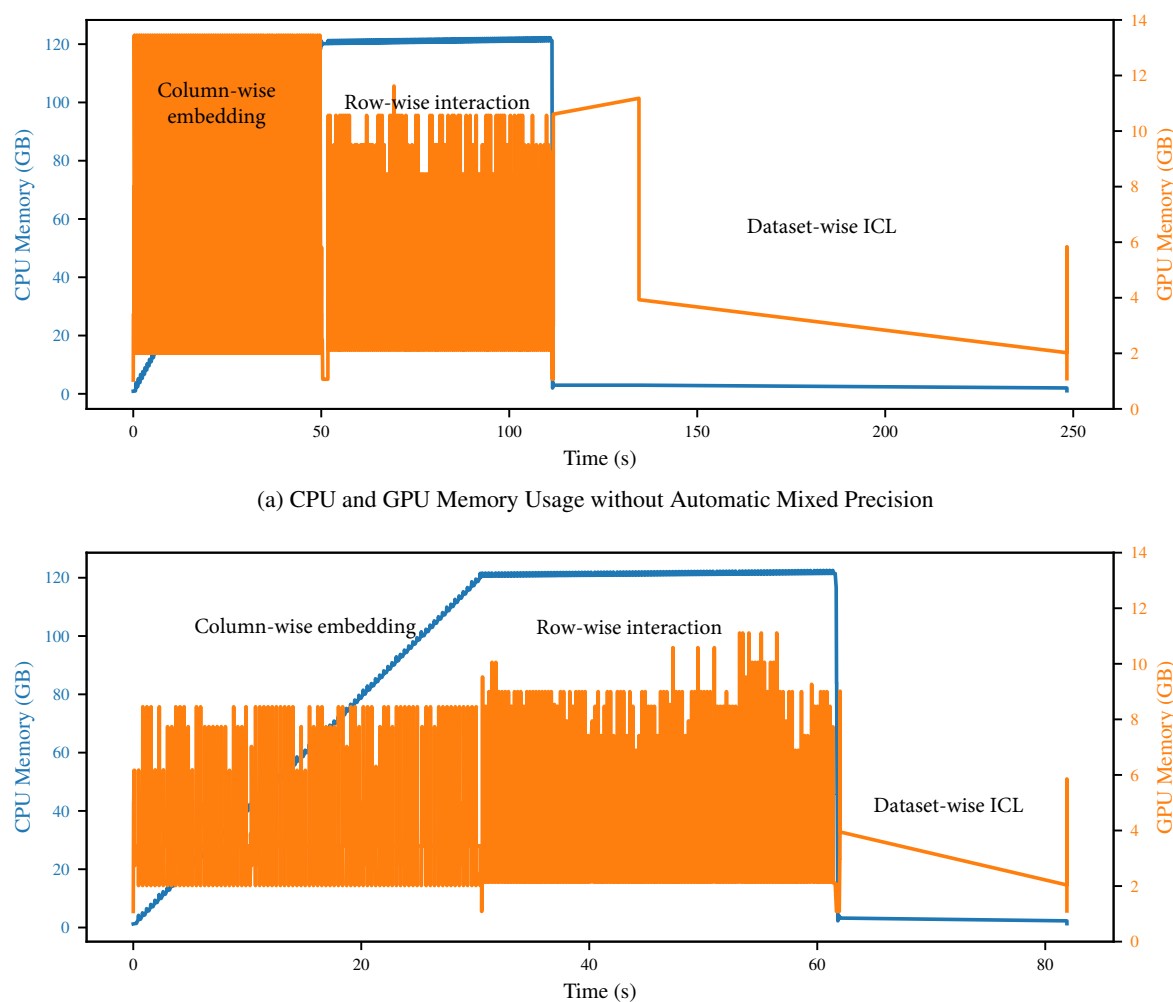

(a) CPU and GPU Memory Usage without Automatic Mixed Precision

(b) CPU and GPU Memory Usage with Automatic Mixed Precision

*Figure E.4.* **CPU and GPU memory consumption during inference for a dataset with 500K samples and 500 features.** The relatively high CPU memory consumption can be largely mitigated by optional disk offloading via memory mapping.

### E.3. Hierarchical Class-extension Strategy

Due to pretraining limitations, TabICL cannot natively handle classification problems with more than 10 classes. To address this, we employ hierarchical classification to extend the class scalability of TabICL. This involves building a multi-level classification tree where:

1. Each internal node splits the classes assigned to it into at most 10 child groups.

2. Recursion stops when a node contains $\leq 10$ classes, turning it into a leaf.

Since each split reduces the class count by at least a factor of 10, the tree depth $r = \left\lceil \log_{10} k \right\rceil$, where $k$ denotes the total number of classes.

Within the tree, each internal node predicts the probabilities of its child groups, while each leaf predicts the probabilities of the classes it contains. For any given class $c$, the unique path from the root to its containing leaf can be represented by a sequence of nodes:

$$g_0(c) = \text{root}, \quad g_1(c), g_2(c), \ldots, g_{r-1}(c), \quad g_r(c) = c$$

where $g_i(c)$ denotes the node at depth $i$ on the path leading to class $c$. Applying the probability chain rule along the path yields the overall posterior probability:

$$p\big(y = c \mid x\big) \;=\; \prod_{i=1}^{r} p\big(g_i(c) \,\big|\, g_{i-1}(c), x, \mathcal{D}_{g_{i-1}(c)}\big) \tag{7}$$

where $x$ is the given test sample, and $\mathcal{D}_{g_{i-1}(c)}$ represents the training data associated with the node $g_{i-1}(c)$. Note that the targets in $\mathcal{D}_{g_{i-1}(c)}$ are the child group indices for the node $g_{i-1}(c)$.

## F. Excluded Development Datasets of TabPFNv2

TabPFNv2 utilized several development datasets for hyperparameter tuning (e.g., learning rate, prior-related parameters), early stopping to prevent overfitting by monitoring performance on these datasets, and model selection to identify the best-performing model among multiple candidates. These development datasets are provided in the supplementary material of TabPFNv2 (Hollmann et al., 2025).

For a fair comparison, we excluded these development datasets from the main paper. In the table below, we report the performance of several high-performing methods on the development datasets included in the TALENT benchmark:

*Table F.1.* Performance on 15 development datasets of TabPFNv2

| Dataset | TabICL | | | TabPFNv2 | | | TabR | | | ModernNCA | | | RealMLP | | | TabM | | | CatBoost | | |
|---|---|---|---|---|---|---|---|---|---|---|---|---|---|---|---|---|---|---|---|---|---|
| | Acc. | AUC | LogL | Acc. | AUC | LogL | Acc. | AUC | LogL | Acc. | AUC | LogL | Acc. | AUC | LogL | Acc. | AUC | LogL | Acc. | AUC | LogL |
| allbp | 0.973 | 0.975 | 0.081 | 0.978 | 0.978 | 0.070 | 0.970 | 0.915 | 0.150 | 0.971 | 0.969 | 0.093 | 0.971 | 0.845 | 0.145 | 0.974 | 0.910 | 0.252 | 0.978 | 0.942 | 0.070 |
| baseball | 0.933 | 0.963 | 0.160 | 0.940 | 0.969 | 0.155 | 0.947 | 0.965 | 0.171 | 0.927 | 0.951 | 0.207 | 0.937 | 0.899 | 0.248 | 0.943 | 0.964 | 0.155 | 0.939 | 0.965 | 0.157 |
| delta_ailerons | 0.952 | 0.981 | 0.161 | 0.954 | 0.982 | 0.159 | 0.948 | 0.978 | 0.175 | 0.949 | 0.980 | 0.165 | 0.945 | 0.976 | 0.184 | 0.950 | 0.979 | 0.172 | 0.948 | 0.979 | 0.164 |
| eye_movements | 0.639 | 0.822 | 0.757 | 0.788 | 0.930 | 0.488 | 0.970 | 0.996 | 0.184 | 0.986 | 0.998 | 0.090 | 0.767 | 0.900 | 0.626 | 0.994 | 1.000 | 0.043 | 0.719 | 0.883 | 0.623 |
| eye_movements_bin | 0.588 | 0.648 | 0.655 | 0.678 | 0.751 | 0.597 | 0.659 | 0.732 | 1.344 | 0.909 | 0.974 | 0.207 | 0.596 | 0.639 | 2.244 | 0.593 | 0.628 | 0.668 | 0.615 | 0.672 | 0.642 |
| heloc | 0.720 | 0.799 | 0.545 | 0.726 | 0.803 | 0.543 | 0.722 | 0.795 | 0.555 | 0.721 | 0.797 | 0.547 | 0.722 | 0.798 | 0.551 | 0.718 | 0.789 | 0.557 | 0.723 | 0.799 | 0.544 |
| hill-valley | 0.785 | 0.917 | 0.518 | 0.983 | 0.992 | 0.098 | 0.700 | 0.732 | 0.571 | 0.781 | 0.804 | 1.455 | 0.796 | 0.852 | 0.547 | 0.512 | 0.529 | 0.703 | 0.517 | 0.515 | 0.693 |
| JapaneseVowels | 0.997 | 1.000 | 0.011 | 0.997 | 1.000 | 0.008 | 0.996 | 1.000 | 0.016 | 0.996 | 1.000 | 0.013 | 0.994 | 1.000 | 0.113 | 0.993 | 1.000 | 0.051 | 0.983 | 1.000 | 0.048 |
| led24 | 0.731 | 0.953 | 0.812 | 0.732 | 0.956 | 0.780 | 0.738 | 0.949 | 0.885 | 0.732 | 0.955 | 0.973 | 0.734 | 0.954 | 0.827 | 0.734 | 0.958 | 0.793 | 0.735 | 0.956 | 0.962 |
| national-longitudinal | 0.998 | 1.000 | 0.005 | 1.000 | 1.000 | 0.000 | 0.994 | 1.000 | 0.029 | 0.999 | 1.000 | 0.003 | 0.995 | 0.996 | 0.064 | 1.000 | 1.000 | 0.000 | 1.000 | 1.000 | 0.012 |
| page-blocks | 0.976 | 0.993 | 0.092 | 0.975 | 0.993 | 0.092 | 0.968 | 0.987 | 0.127 | 0.967 | 0.989 | 0.129 | 0.971 | 0.980 | 0.162 | 0.968 | 0.985 | 0.127 | 0.967 | 0.991 | 0.110 |
| ringnorm | 0.980 | 0.997 | 0.066 | 0.980 | 0.997 | 0.055 | 0.980 | 0.995 | 0.075 | 0.980 | 0.996 | 0.072 | 0.976 | 0.996 | 0.079 | 0.980 | 0.997 | 0.062 | 0.971 | 0.995 | 0.083 |
| rl | 0.763 | 0.853 | 0.472 | 0.860 | 0.937 | 0.322 | 0.878 | 0.934 | 0.334 | 0.837 | 0.916 | 0.378 | 0.747 | 0.824 | 0.548 | 0.787 | 0.869 | 0.552 | 0.786 | 0.863 | 0.463 |
| thyroid-ann | 0.987 | 0.999 | 0.028 | 0.995 | 1.000 | 0.017 | 0.986 | 0.996 | 0.068 | 0.987 | 0.996 | 0.088 | 0.992 | 0.998 | 0.087 | 0.992 | 1.000 | 0.024 | 0.995 | 1.000 | 0.012 |
| waveform-5000 | 0.872 | 0.978 | 0.259 | 0.865 | 0.977 | 0.263 | 0.860 | 0.975 | 0.310 | 0.862 | 0.975 | 0.278 | 0.867 | 0.975 | 0.291 | 0.862 | 0.976 | 0.279 | 0.860 | 0.973 | 0.297 |

## G. Random Forest Extension of TabPFNv2 for Large Datasets

Hollmann et al. (2025) proposed an extension of TabPFNv2 using random forest (RF) to enhance its performance on large datasets. During training, the extension recursively partitions the feature space using standard decision tree splitting criteria.

At inference time, instead of relying on simple majority voting at the leaves, it fits TabPFNv2 to the subset of training data that reaches each leaf. This extension not only leverages the inductive biases of tree-based models but also reduces the number of samples processed by TabPFNv2, thereby mitigating its scalability limitations.

The ensemble prediction of this RF extension involves two main steps: (1) routing each test sample through all trees to their corresponding leaf nodes, and (2) aggregating the predictions from TabPFNv2 at these leaves by either averaging probabilities or logits.

A key innovation of this RF extension is the adaptive pruning mechanism that dynamically determines whether to use parent node predictions or leaf node predictions based on validation performance. This prevents overfitting and optimizes the tree depth for each region of the feature space.

This extension is not exclusive to TabPFNv2 and can also be applied to other tabular foundation models like TabICL. We evaluated this RF extension with adaptive pruning enabled for both TabPFNv2 and TabICL on large datasets with more than 10K samples. Additionally, we set the ensemble size for both TabPFNv2 and TabICL to 4, considering that random forest already ensembles multiple decision trees. As shown in Figure G.1, the RF extension significantly improves the performance of both TabPFNv2 and TabICL.

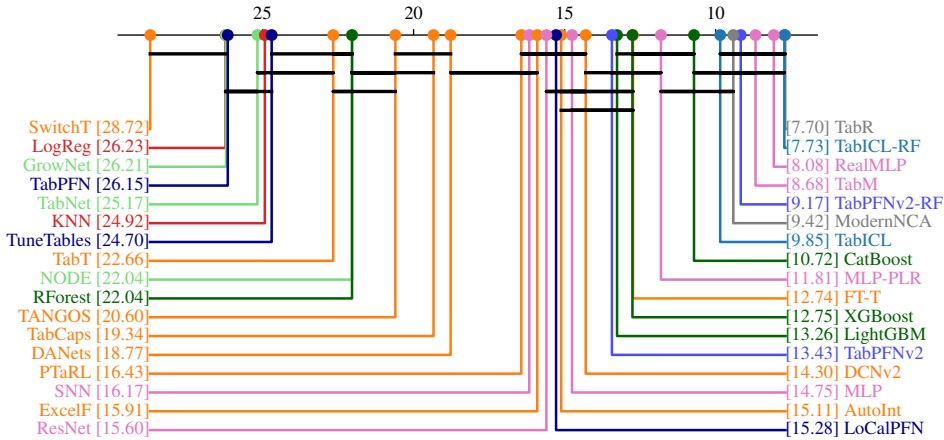

*Figure G.1.* **Critical difference diagram (based on accuracy) on 53 large datasets with more than 10K samples.** For TabICL-RF and TabPFNv2-RF, the "RF" suffix indicates the use of the random forest extension.

## H. Effect of Ensemble Size on TabPFNv2 and TabICL

Both TabPFNv2 and TabICL break column-order invariance to mitigate the learning collapse problem, as described in Section 3.3. To approximately restore permutation invariance, they aggregate predictions across multiple column permutations, while also using different preprocessors and shuffling class labels. We analyze the effect of ensemble size on their performance, as shown in Figure H.1. Our findings indicate that TabPFNv2 shows a clear improvement even with a small ensemble size ($\leq 4$), whereas TabICL requires a larger ensemble size ($\geq 8$) for a noticeable performance gain.

Overall, TabPFNv2 benefits more from ensembling than TabICL. This could be attributed to :

- TabPFNv2 employs more sophisticated preprocessing operations compared to TabICL.

- TabPFNv2 benefits more from column permutations inherently. Unlike TabICL, TabPFNv2 never collapses the column dimension. This means permuting columns could have a more pronounced impact on the outputs of TabPFNv2, which reduces the correlation between ensemble members of TabPFNv2, thereby improving the effectiveness of ensembling.

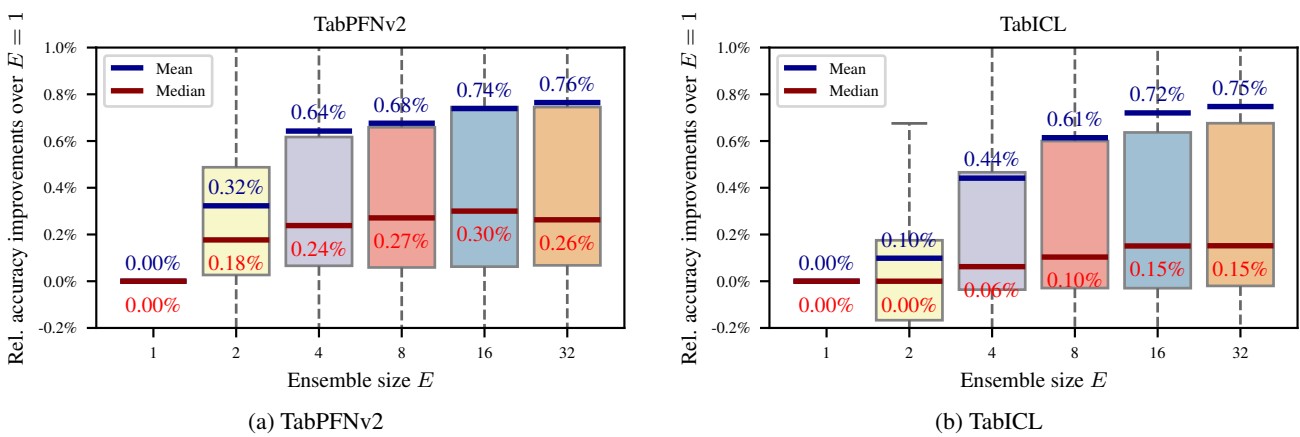

*Figure H.1.* **Effect of ensemble size on TabPFNv2 and TabICL.** The figures show the relative accuracy improvements of different ensemble sizes over no ensembling across 171 datasets with at most 10 classes.

## I. Average Performance and Rankings

In this section, we present the average rank of all methods across different dataset categories, including binary datasets, multi-class datasets (≤10 classes), small classification datasets (≤10K samples), and large classification datasets (>10K samples). The rankings are computed based on accuracy, AUC, and Log Loss. The average rank is given in critical difference diagrams, with Wilcoxon-Holm tests with a significance level 0.05. The lower the rank value, the better the performance.

It is important to note that accuracy is used as the objective metric for hyperparameter tuning in tabular methods. Moreover, LoCalPFN (Thomas et al., 2024) and TuneTables (Feuer et al., 2024) are only included in the accuracy-based rankings.

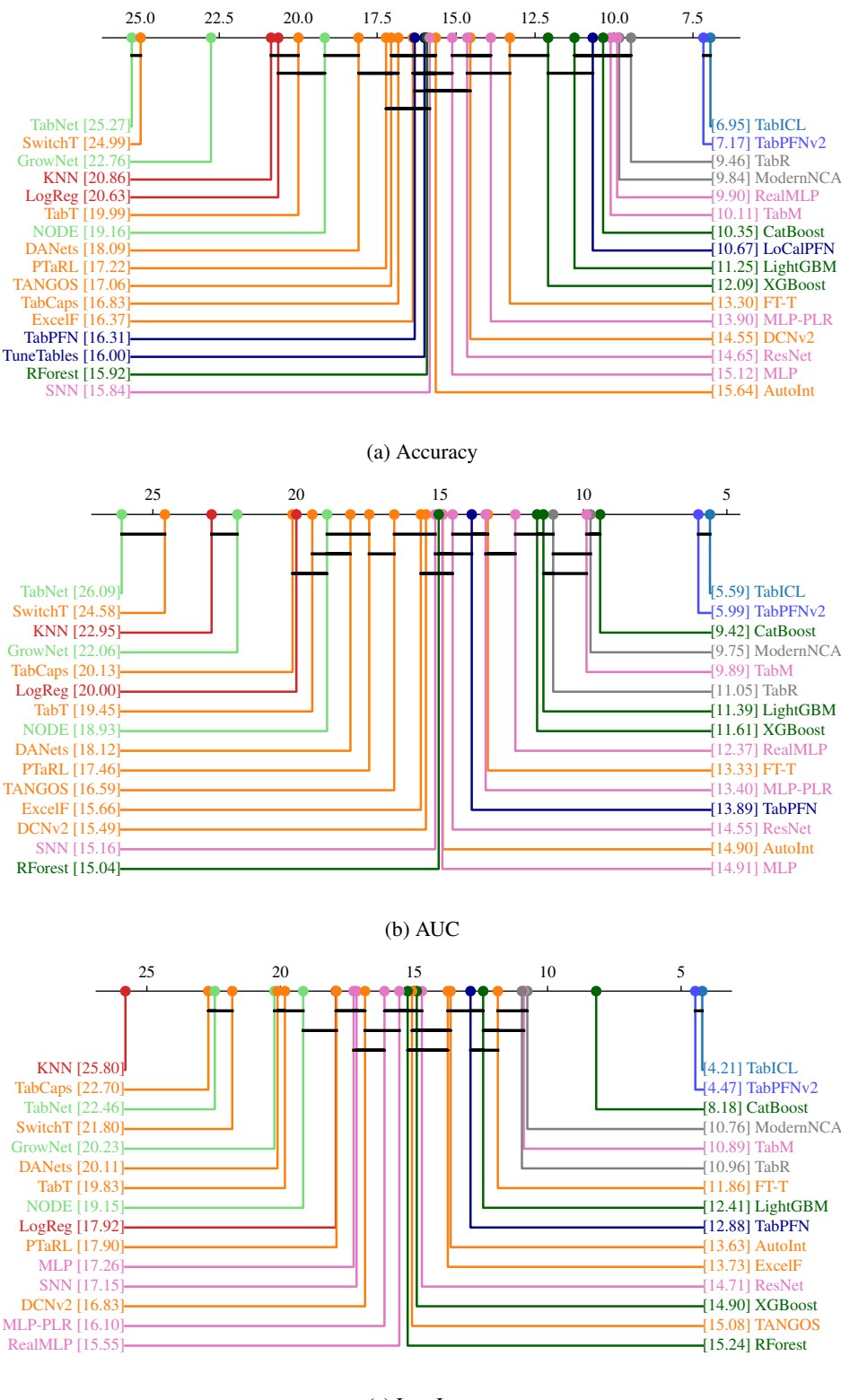

(a) Accuracy

(b) AUC

(c) Log Loss

*Figure I.1.* **Critical difference diagram for all 171 classification datasets ($\leq$ 10 classes).**

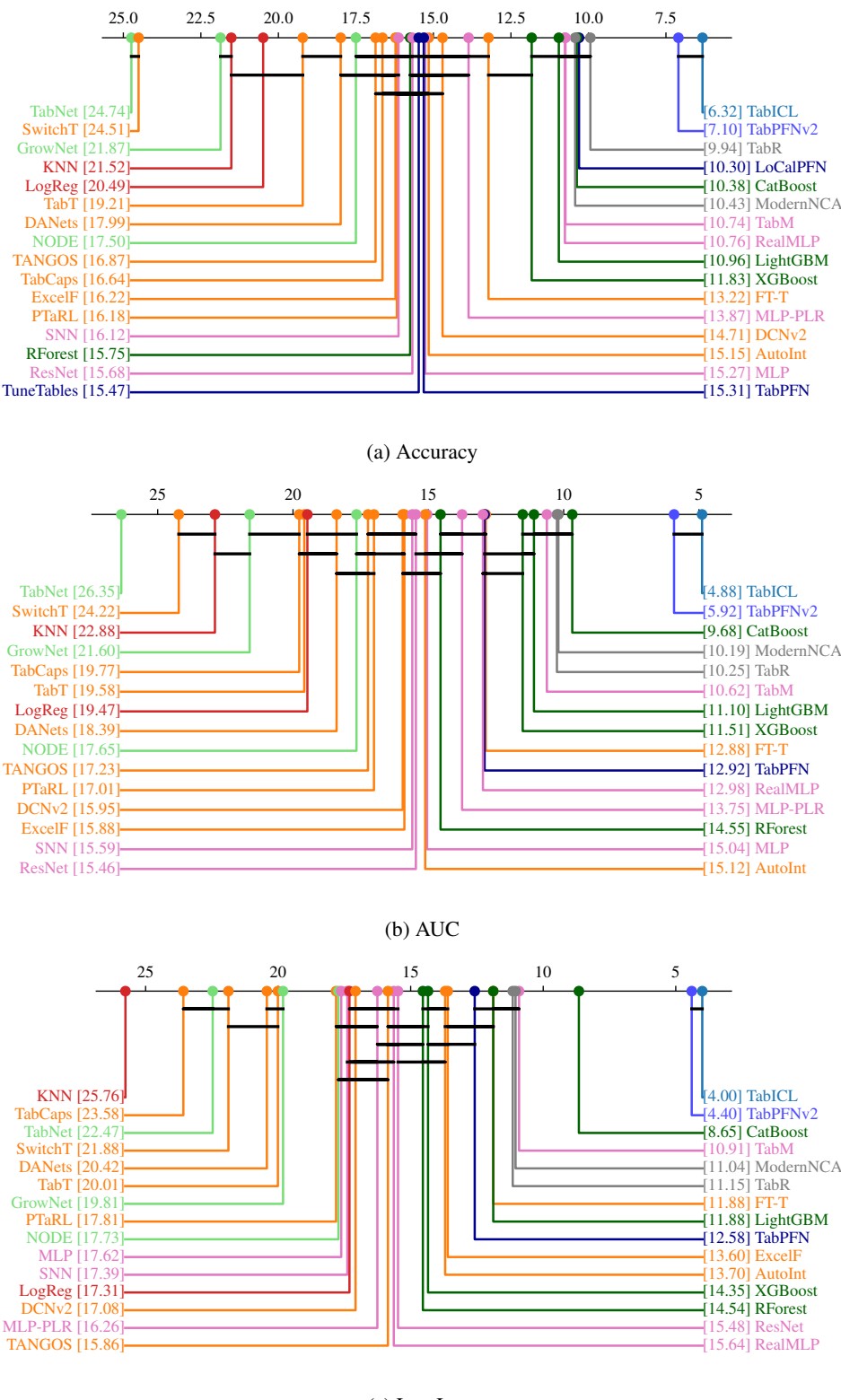

(a) Accuracy

(b) AUC

(c) Log Loss

*Figure I.2.* **Critical difference diagram for 112 binary classification datasets.**

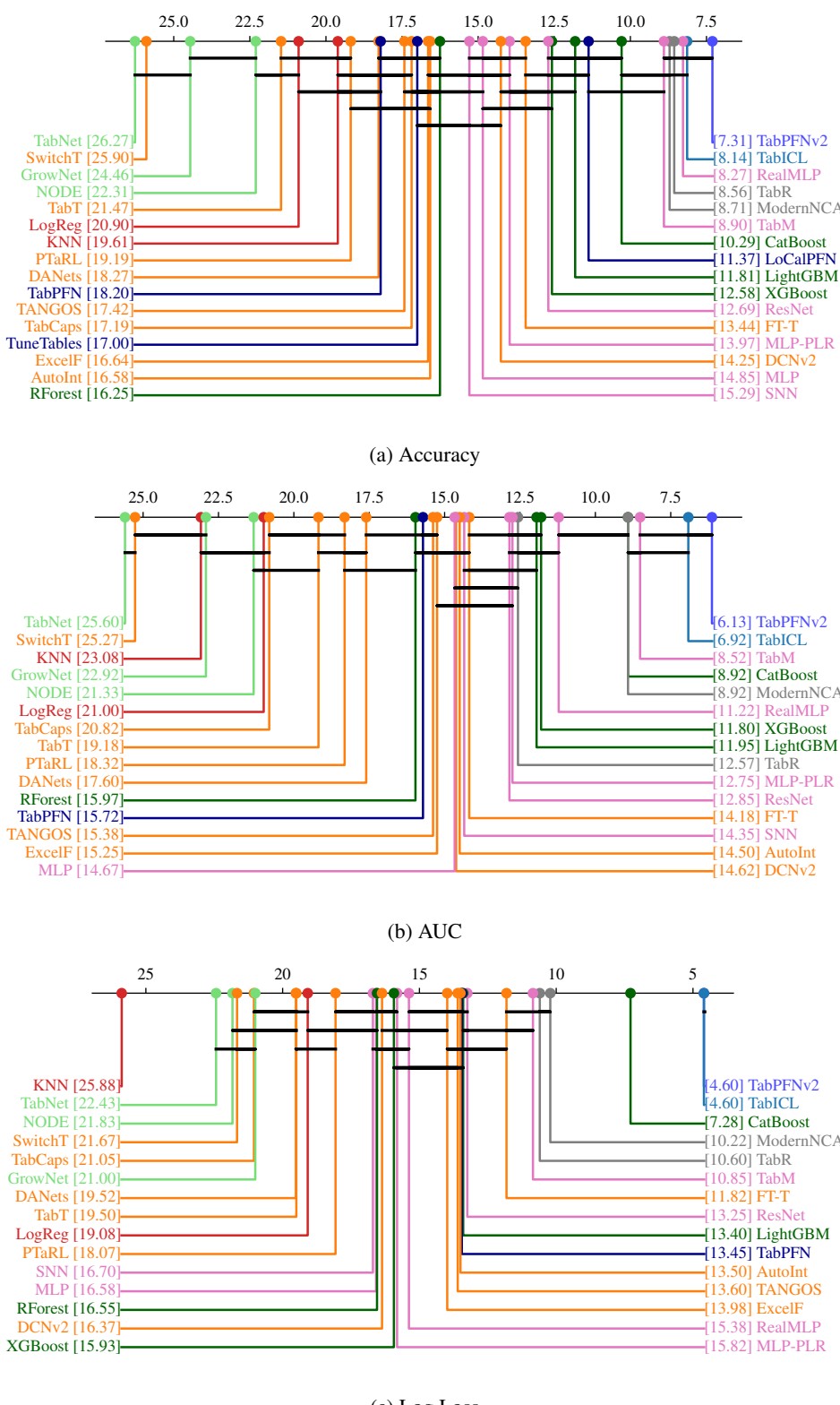

(a) Accuracy

(b) AUC

(c) Log Loss

*Figure I.3.* **Critical difference diagram for 59 multi-class classification datasets ($\leq$ 10 classes).**

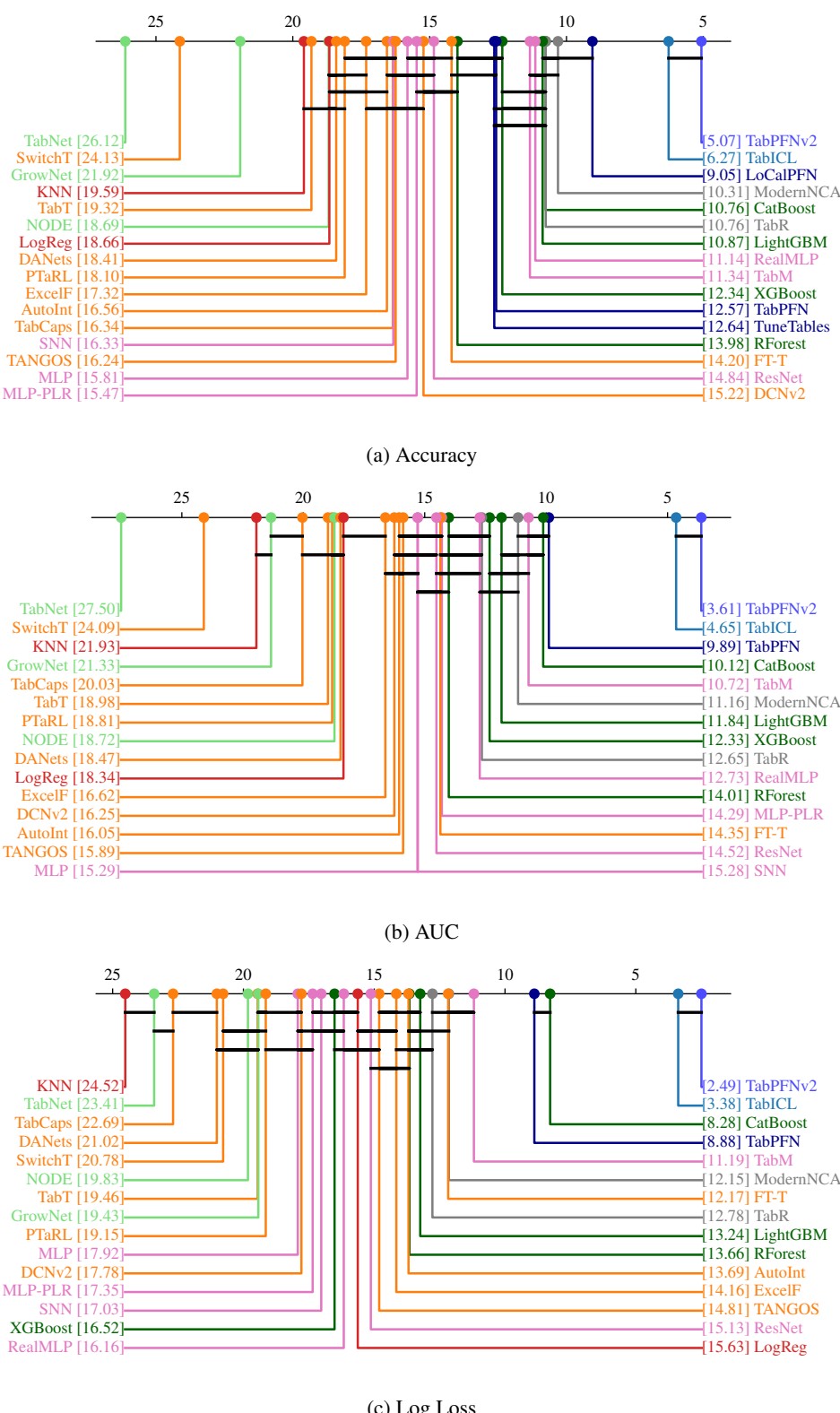

(a) Accuracy

(b) AUC

(c) Log Loss

*Figure I.4.* **Critical difference diagram for 116 small classification datasets (≤ 10K samples).**

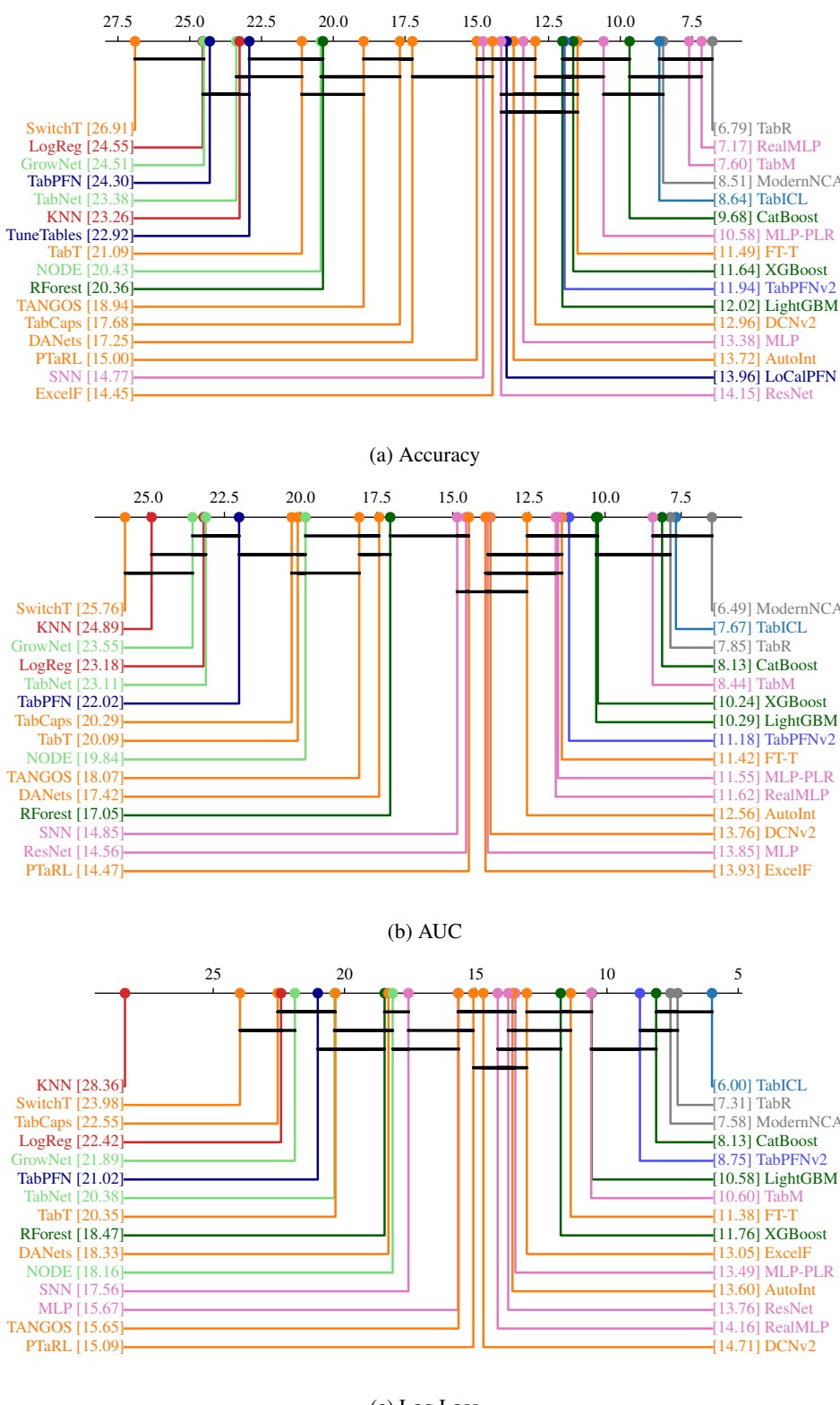

(a) Accuracy

(b) AUC

(c) Log Loss

*Figure I.5.* **Critical difference diagram for 53 large classification datasets ($>$ 10K samples).**

