# OpenReview forum: "TabICL: A Tabular Foundation Model for In-Context Learning on Large Data"
_ICML.cc/2025/Conference — ICML 2025 poster_

### Official Review · Reviewer_VxC8 · 2025-03-04

**Overall Recommendation:** 4

**Summary:**

This paper propose a Tabular Foundation Model (TabICL) for classification tasks with new Distribution-aware Column-wise Embedding then Context-aware Row-wise Interaction design. It is a scalable and efficient model,  it can handle up to 500K samples and 500 features with around 20GB of GPU memory. This model demonstrate the SoTA level performance on multiple benchmarks.

**Claims And Evidence:**

Yes

**Essential References Not Discussed:**

N/A

**Experimental Designs Or Analyses:**

Yes

**Methods And Evaluation Criteria:**

Yes

**Other Comments Or Suggestions:**

Just a few more questions regarding to some details, see questions section

**Other Strengths And Weaknesses:**

- This model has significant better efficiency compared with the prior works, handling up to 500K samples and 500 features while maintaining reasonable computational costs. It demonstrates its potentials on the large datasets.


- The model is fully evaluated with its accuracy and efficiency on the latest dataset, showing strong performance compared with other methods.


- However, it would be great if we can see more comparison results against the more recent models based on TabPFN. List some related papers below,


[1] Feuer, Benjamin, Robin Tibor Schirrmeister, Valeriia Cherepanova, Chinmay Hegde, Frank Hutter, Micah Goldblum, Niv Cohen, and Colin White. "TuneTables: Context Optimization for Scalable Prior-Data Fitted Networks." CoRR (2024).

[2] Ye, Han-Jia, Si-Yang Liu, and Wei-Lun Chao. "A Closer Look at TabPFN v2: Strength, Limitation, and Extension." arXiv preprint arXiv:2502.17361 (2025).

[3] Xu, Derek Qiang, F. Olcay Cirit, Reza Asadi, Yizhou Sun, and Wei Wang. "Mixture of In-Context Prompters for Tabular PFNs." In The Thirteenth International Conference on Learning Representations.

[4] Thomas, Valentin, Junwei Ma, Rasa Hosseinzadeh, Keyvan Golestan, Guangwei Yu, Maks Volkovs, and Anthony L. Caterini. "Retrieval & fine-tuning for in-context tabular models." Advances in Neural Information Processing Systems 37 (2025): 108439-108467.

**Questions For Authors:**

- Can you provide a brief comparison between TabPFNv2 and your model on the method aspect? It would be great if it can be provided in the related work section.


- Can you simply explain what's the benefit of using Set Transformer compared with the weight sharing? How much computation overhead it can bring and how much it makes the performance better?


- Can you explain why diversifying activation functions (15 additional activation functions) can help the training? Besides, how you perform the random rescaling?

**Relation To Broader Scientific Literature:**

The prior SoTA method's architecture limits scalability for large datasets, and this paper's method is more scalable and computationally efficient.

**Theoretical Claims:**

N/A

---

> ### Author Rebuttal · Authors · 2025-03-31
>
> We thank the reviewer for their insightful review and recognition of our work.
>
> ---
> 1. *More comparison results against the more recent models based on TabPFN ?*
>
> Thanks for these relevant works. Due to limited computational resources, we prioritized methods based on code availability.
> - TuneTables: Official code is available.
> - A Closer Look at TabPFNv2: No official code was available at the time of evaluation. Moreover, the paper was released online after the ICML submission deadline.
> - MixturePFN (Mixture of In-Context Prompters): No code was publicly available.
> - LocalPFN (Retrieval & fine-tuning for in-context tabular models): Code is found.
>
> Based on these considerations, we decided to evaluate TuneTables and LocalPFN across 186 datasets with $\le 10$ classes on the TALENT benchmark. Their average ranks (based on accuracy), along with other methods, are summarized below:
>
> | Model       | Avg. Rank |
> |-------------|------------|
> | TabICL      | 6.60       |
> | TabPFNv2    | 6.95       |
> | TabR        | 9.04       |
> | RealMLP     | 9.45       |
> | ModernNCA   | 9.49       |
> | CatBoost    | 9.94       |
> | LocalPFN    | 10.26      |
> | LightGBM    | 10.80      |
> | XGBoost     | 11.60      |
> | FT-T        | 12.74      |
> | MLP-PLR     | 13.35      |
> | DCNv2       | 14.02      |
> | ResNet      | 14.09      |
> | MLP         | 14.58      |
> | AutoInt     | 15.05      |
> | SNN        | 15.26      |
> | RForest    | 15.38      |
> | TuneTables | 15.47      |
> | TabPFN     | 15.74      |
>
> ---
>
> 2. *A brief comparison between TabPFNv2 and your model on the method aspect ?*
>
> In the updated version, we will include a methodological comparison between TabPFNv2 and TabICL in the related work section. Below is a brief summary:
> | Aspect | TabPFNv2 | TabICL |
> | --- | --- | --- |
> | Attention | Alternating column / row attentions | Column→Row + In-context Learning |
> | Label Fusion | Early (input layer) | Late (during ICL) |
> | Collapse Issue | Random feature embeddings and feature grouping | Rotary positional embedding |
> | Scalability  | ≤10K samples, ≤10 classes | 100K samples, any class |
>
> ---
>
> 3. *The benefit of using Set Transformer compared with the weight sharing ? ...*
>
> We admit that we may not fully understand the question. We guess that this question may refer to Lines 194–196 in our paper: *"Instead of sharing the same weight and bias for all samples in a column as in a classic linear layer, each value is projected using its own weight and bias."*
>
> During column-wise embedding (feature embedding), we map each individual cell in a column $c_j$ into an embedding vector. Specifically, we compute:
>
> $W, B = \text{SetTransformer}(c_j)$
> $e_j = W \odot c_j + B$
>
> Here, we consider two choices:
> - Shared weights and biases: All cells in column $c_j$ share the same weight and bias
> - Cell-specific weights and biases (our choice): Each cell $c_j^i$  is associated with its own weight $W_i$ and bias $B_i$.
>
> We opt for the second design because, in tabular data, cells within the same column often serve distinct statistical roles—e.g., being the minimum, maximum, mode, or an outlier. Using cell-specific parameters allows the model to adaptively embed each value based on such roles.
>
> Importantly, both options rely on the set transformer to generate the projection parameters, so the architectural backbone remains the same. The key difference lies in how the output of the set transformer is used.
>
>
> *How much computation overhead it can bring ?*
>
> Using separate weights and biases for each cell introduces only a mild overhead, as the set transformer still processes the entire column as a set, and the per-cell projections are implemented efficiently as element-wise operations.
>
> *How much it makes the performance better ?*
>
> In our early experiments, we observed that using per-cell weights and biases improved performance over shared ones. However, we acknowledge that we have not conducted a thorough ablation study to quantify this gain precisely, and we will consider including such analysis in future work.
>
> ---
>
> 4. *Why diversifying activation functions can help the training? How you perform the random rescaling ?*
>
> Our intuition for diversifying activation functions is that it is generally helpful to increase the diversity of generated datasets. In particular, some of the activation functions can generate functions of different smoothness (e.g., the GP functions), periodic functions (sine), (semi-)discrete feature distributions (e.g. ReLU/signum), and heavy-tailed distributions (e.g. exp). We will add this to Appendix B.1. The random rescaling is already described in Appendix B.1: $x_i \leftarrow \exp(2a)(x_i + b)$ with $a, b \sim \mathcal{N}(0, 1)$.
>
> ---
>
> **In the revised version, we will include LocalPFN and TuneTables into the benchmark.**

---

> > ### Comment · Reviewer_VxC8 · 2025-04-03
> >
> > Thanks for the clarification! Most of my concerns are well addressed.

---

### Official Review · Reviewer_gpuG · 2025-03-11

**Overall Recommendation:** 4

**Summary:**

The paper proposes TabICL a tabular foundation model able to fit larger datasets than TabPFN (v1) and TabPFN (v2).
The main difference lies in the embedding strategy which leverages Set-Transformer as well as later in the model a positional rotary embedding layer. Similar to TabPFN v2 using FlashAttention, makes the model more efficient than v1.
These adaptations as well as fewer ow-wise column-wise attentionlayers and smaller dimensions make the model also more efficient than TabPFN v2 and allow it to be fit on larger datasets.

**Claims And Evidence:**

- I am not quite convinced by the positional invariance claims. First, the row-wise attention achieves positional invariance only to later again lose it by introducing the positional rotary embeddings. As far as I understand the architecture, it is not truly positionally invariant and does not distinguish itself in that regard from TabPFN v1. Could you please clarify?
  - Given that TabPFNv2 does achieve the invariance via row-col attention

- A major drawback is, that the claims and evidence are very hard to test, since neither the datasets nor the code or the model was released upon submission.

**Essential References Not Discussed:**

- All essential references are discussed. However, there are a few minor and fairly new references that should be included

---
**Models**
[1] Gorishniy, Yury, Akim Kotelnikov, and Artem Babenko. "Tabm: Advancing tabular deep learning with parameter-efficient ensembling." arXiv preprint arXiv:2410.24210 (2024).
[2] Thielmann, Anton Frederik, et al. "Mambular: A sequential model for tabular deep learning." arXiv preprint arXiv:2408.06291 (2024).
[3] den Breejen, Felix, and Se-Young Yun. "Attic: A New Architecture for Tabular In-Context Learning Transformers."

**Theory**
[4] Reuter, Arik, et al. "Can Transformers Learn Full Bayesian Inference in Context?." arXiv preprint arXiv:2501.16825 (2025).

**Experimental Designs Or Analyses:**

- As described above, I am familiar with the benchmark, and all the results align with the original. However, since nothing was released yet from the authors it is hard to verify the soundness of the presented results.

**Methods And Evaluation Criteria:**

- Yes the method and evaluation criteria seem correct. However, as indicated above they are not reproducible.
- There are a few newer DL models [1, 2] missing from the benchmarks as well as ICL models [3]


---
[1] Gorishniy, Yury, Akim Kotelnikov, and Artem Babenko. "Tabm: Advancing tabular deep learning with parameter-efficient ensembling." arXiv preprint arXiv:2410.24210 (2024).
[2] Thielmann, Anton Frederik, et al. "Mambular: A sequential model for tabular deep learning." arXiv preprint arXiv:2408.06291 (2024).
[3] den Breejen, Felix, and Se-Young Yun. "Attic: A New Architecture for Tabular In-Context Learning Transformers."

**Other Comments Or Suggestions:**

- line 257 "probably" -> "probable / likely"
- line 295 "gies" -> "gives"

**Other Strengths And Weaknesses:**

- The results are really great and the performed benchmark is the best available benchmark that one could use.
The majore problem with this is, that nothing was released. That TabPFN v2 did not release the training datasets is already a shame and I hope the field does not move to a direction where models are being kept private. I do believe that there is a place for this contribution in ICML (given the authors incorporate/answer the reviews) but **only** if the datasets, model weights and training info are published. Since the authors did not submit an anonymous version of this, my current evaluation is rather strict.

**Questions For Authors:**

- Why exactly do you use the 4 cls tokens? Do you also have results for fewer/more cls tokens?
- How do you explain that the rotary positional embeddings can mitigate the embedding collapse problem? Since you want to achieve positional invariance, how does including the feature position help in distributional disentanglement?
- Have you also tried out other invariance strategies, i.e. convolutions [1].


---
[1] Thielmann, Anton Frederik, et al. "Mambular: A sequential model for tabular deep learning." arXiv preprint arXiv:2408.06291 (2024).

**Relation To Broader Scientific Literature:**

- The paper fits well within the emerging literature of tabular ICL models. Given the recent publication of TabPFN v2, the contributions do seem a little minor and more in the direction of combining concepts together, to squeeze performance (SetTransformer [1] + Rotary [2] Embedding + Tree based priors [3]).

---
[1] Lee, Juho, et al. "Set transformer: A framework for attention-based permutation-invariant neural networks." International conference on machine learning. PMLR, 2019.
[2] Su, Jianlin, et al. "Roformer: Enhanced transformer with rotary position embedding." Neurocomputing 568 (2024): 127063.
[3] Breejen, Felix den, et al. "Fine-tuned In-Context Learning Transformers are Excellent Tabular Data Classifiers." arXiv preprint arXiv:2405.13396 (2024).

**Theoretical Claims:**

- There are no theoretical proofs.
- However, the paper could benefit from validating some of the effects of the introduced methods. For instance one of the core claims is that via the SetTransformer, positional invariance is achieved, however, what exactly is the benefit of the SetTransformer, given the Rotary Positional embedding layer?

---

> ### Author Rebuttal · Authors · 2025-03-31
>
> We thank the reviewer for their insightful review, and for mentioning that “the results are really great”.
>
> Next, we respond to the reviewer's questions.
>
> ---
> 1. *A major drawback is ... the code or the model was released upon submission*
>
> We **confirm that all of TabICL will be made fully public, including model weights, training details and prior data generation code**. We have already released the inference code after consolidating it to a python package with a fully scikit-learn compliant interface. Here is an anonymous version: https://anonymous.4open.science/r/tabicl-A70B or download code from https://limewire.com/d/opefZ#jPuHwDXBUg. The pre-training code and data generation will be released upon acceptance.
>
> We agree with the reviewer and believe open-sourcing our work will significantly boost its impact—especially since TabPFNv2 remains partially closed, which limits the community’s ability to reproduce or build upon it.
>
> ---
>
> 2. *I am not quite convinced by the positional invariance claims...*
>
> Neither TabPFNv1, TabPFNv2, nor TabICL achieves permutation invariance of columns. All these models need to break this invariance in some way to solve the learning collapse issue when some features have similar statistics:
> - TabPFNv1 relies on fixed, feature-specific embedding layers to implicitly encode column identities.
> - TabPFNv2 introduces random feature embeddings and feature grouping.
> - Our TabICL adopts RoPE.
>
> We do not claim permutation invariance; Instead, we just address the issue differently from the other two models.
>
> ---
>
> 3. *There are a few newer DL models [1, 2] missing ...*
>
> Thanks for these relevant works. Due to limited computational resources, we prioritized methods based on their reported performance and availability.
>
> - TabM has been accepted at ICLR and showed promising results across 46 datasets. Its code is open-source, so we evaluated TabM on TALENT.
>
> - Mambular shows equal performance compared to CatBoost on 12 datasets, according to both its ICLR rebuttal and the latest arXiv version, so we have deprioritized it for now.
>
> - Attic does not provide publicly available code, is not on arXiv and was only visible on OpenReview (submitted and now rejected from ICLR).
>
> TabM performance rank is as follows:
> | Model | Avg. Rank |
> |---|---|
> | TabICL|6.48|
> | TabPFNv2| 6.84|
> | TabR|8.87|
> | RealMLP | 9.24|
> | ModernNCA | 9.30|
> | TabM | 9.79|
> | CatBoost| 9.93|
> | LightGBM| 10.57|
> | XGBoost|11.51|
>
> ---
> 4. *the benefit of the set transformer given RoPE ?*
>
> We’re unsure we fully understand the question. The set transformer is used for column-wise embedding and is *row-order* invariant, while RoPE later breaks *column-order* invariance to mitigate learning collapse. They are not directly related.
>
> We chose the set transformer for its linear complexity with respect to sequence length (see response to Reviewer rh11, Q3), which is more efficient than standard self-attention—especially on large tabular datasets.
>
> ---
>
> 5. *Given the recent TabPFNv2, the contributions do seem [...] combining ...*
>
> We would like to emphasize that **TabICL was designed, developed, and pretrained independently of TabPFNv2**. TabPFNv2 was published very shortly before ICML submission deadline, leaving little time to include it as a baseline. However, we still rigorously compared against it in our paper.
>
> TabICL and TabPFNv2 differ in many ways:
> | Aspect | TabPFNv2 | TabICL |
> | --- | --- | --- |
> | Attention | Alternating column / row attentions | Column→Row + ICL |
> | Label Fusion | Early (input layer) | Late (during ICL) |
> | Collapse Issue | Feature grouping | RoPE |
> | Scalability  | ≤10K samples, ≤10 classes | 100K samples, any class |
>
> Additional innovations in TabICL:
> - Hierarchical classification for >10 classes
> - Curriculum pretraining (1K→60K samples)
> - Memory optimizations (5GB GPU for 100K samples)
>
> The reviewer questions whether these contributions are "minor", but the empirical results demonstrate otherwise: TabICL is faster, more scalable, and also delivers better performance.
>
> ---
>
> 6. *New references that should be included*
> Thanks for these recent references. We will include them.
>
> ---
>
> 7. *Why exactly do you use the 4 cls tokens ?*
>
> The choice of 4 [CLS] tokens was heuristic—we considered it a reasonable trade-off but did not conduct ablation studies with fewer or more [CLS] tokens.
>
> ---
>
> 8. *How RoPE can mitigate the embedding collapse problem ?*
>
> RoPE mitigates embedding collapse by breaking feature permutation symmetry. After column-wise embedding, features with similar distributions may have indistinguishable embeddings. RoPE injects positional information via rotational encoding for row-wise interaction, ensuring distinct representations even for permuted columns. This is validated in Figure 4 (Section 3.3).
>
> ---
>
> 9. *Other invariance strategies ?*
>
> Yes, we tried CNN to process feature embeddings after column-wise embedding, but RoPE is better.
>
> ---
>
>
> **In the revised version, we will include TabM.**

---

> > ### Comment · Reviewer_gpuG · 2025-04-01
> >
> > # Answer 1:
> >
> > Dear authors,
> >
> > thank you for your answers and comments.
> >
> > Since the major point - publicly available code is resolved (thank you for that!) - I increased my score 2-> 3.
> >
> > I do have a few other questions to clarify my understanding:
> >
> > > We do not claim permutation invariance; Instead, we just address the issue differently from the other two models.
> > - Thats not really correct. you quite prominently claim positional invariance:
> > - > Unlike the architecture of TabPFNv1, the row-wise attention in TabICL allows to incorporate this invariance naturally
> >
> > - Could you clarify?
> > - And that is what my question regarding SetTransformer and the Rotary embedding is about: As far as I undestand your manuscript, you achieve positional invariance via the SetTransformer. However, positional invariance is lost again due to the Rotary embeddings. Is that correct?
> >
> >
> > - Thanks for including TabM, I wasn't aware that Attic is not publicly available: The more so I appreciate, your code release!
> > - This will not impact my evaluation, and I understand the computational costs. However, if feasible, I would appreciate seeing Mambular included in the benchmark. Its positional invariance claims and autoregressive structure make for an interesting comparison to TabICL.
> >
> >
> > Thanks for clearly answering the other questions.
> >
> >
> > # Answer 2.
> >
> > Dear authors, since I can only give one answer, I edited this one.
> > Thanks for clarifying my misunderstanding on the positional invariance.
> > I increased the score accordingly:
> > 3 -> 4

---

> > > ### Author Response · Authors · 2025-04-02
> > >
> > > Dear Reviewer,
> > >
> > > We sincerely appreciate your active engagement and your recognition of our efforts to open-source the code—and of course, for increasing the score ! Please find our responses to your questions below.
> > >
> > > ---
> > > 1. *... Could you clarify?*
> > >
> > > Thanks for pointing this out. We acknowledge that the sentence *“Unlike TabPFNv1, the row-wise attention in TabICL allows to incorporate this invariance naturally”* was inappropriate and could misleadingly suggest that TabICL achieves invariance to column order. We will remove this statement in the revised version (specifically L239–243).
> > >
> > > The [Why Tabular Foundation Models Should Be a Research Priority] put forward that column-order invariance is a desirable property for tabular foundation models. However, TabPFNv1, TabPFNv2, and TabICL all break this due to the learning collapse issue when features have similar distributional statistics.
> > >
> > > The nature paper of TabPFNv2 wrote:
> > > - *To allow our model to differentiate features more easily that have the same statistics, for example, two features that have the same entries just in different orders, we use random feature embeddings that we add to all embeddings before the first layer.*
> > > - *We found that encoding groups of features can be even more effective compared with encoding one value per representation.*
> > >
> > > In contrast to random feature embeddings and feature grouping, TabICL uses rotary positional embedding (RoPE) to tackle this problem. In fact, both TabPFNv2 and TabICL prioritize empirical performance over strict column-order invariance. Of course, we view this break of column-order invariance as a shared limitation of tabular foundation models, including at least TabPFNv1, TabPFNv2 and TabICL. How to avoid learning collapse while preserving column-order invariance is a key direction for our future work.
> > >
> > > ---
> > >
> > > 2. *... Is that correct?*
> > >
> > > To clarify, the Set Transformer $TF_{col}$ applies attention across all samples within each column during column-wise embedding, while another transformer $TF_{row}$ applies attention across all features within each row during row-wise interaction.
> > >
> > > As mentioned in our previous response, rotary positional embedding (RoPE) does break column-order invariance in order to address the learning collapse problem and improve empirical performance. Without RoPE, TabICL is invariant to column order for the following reasons:
> > >
> > > - $TF_{col}$ processes each feature (column) independently, so changing the order of columns does not affect its output.
> > > - Without RoPE or any other positional encoding, attention is a permutation-invariant operation. Thus, $TF_{row}$ treats each row as a set of features without considering their order, and is inherently invariant to column permutations.
> > >
> > > ---
> > >
> > > 3. *... If feasible, I would appreciate seeing Mambular included in the benchmark.*
> > >
> > > Thank you for your understanding regarding computational resources. We agree that Mambular, with its Mamba-based autoregressive structure, offers an interesting comparison to attention-based models like TabPFNv2 and TabICL.
> > >
> > > Thanks to its open-source code at https://github.com/basf/mamba-tabular, we are actively looking into it. However, integrating it into our benchmark is not easy at the moment. First, the Mambular paper did not perform hyperparameter tuning and provide a suggested search space, which makes fair comparison challenging. Second, we are also limited by computational resources currently. However, we will continue to follow its progress and consider incorporating it in future work.
> > >
> > > ---
> > >
> > > Once again, we sincerely thank you for increasing the score. If you have any further questions or suggestions, please don’t hesitate to let us know—we will do our best to address them.

---

### Official Review · Reviewer_ba5G · 2025-03-12

**Overall Recommendation:** 4

**Summary:**

This paper introduces a tabular foundation model, TabICL, to deal with the classification of larger number of table samples at affordable resources. Compred to the recent state-of-the-art TabPFNv2, the proposed TabICL has competitive performance while enjoys up to 10 times faster speed. The proposed architecture mainly involves three components, which are column-wise transformer for feature embedding, row-wise transformer module for rows interaction, and an additional transformer module for in-context learning. During feature embedding stage, the module is built on top of set transformer, which can capture distribution-aware features. During row features interaction stage, rotary position encoding is used to avoid mode collapse. The experimental results on TALENT benchmark shows the effectiveness of the proposed foundation model.

**Claims And Evidence:**

yes

**Essential References Not Discussed:**

no

**Experimental Designs Or Analyses:**

yes

**Methods And Evaluation Criteria:**

yes

**Other Comments Or Suggestions:**

1. Suggest to indicate that the red bar is median relative accuracy in Fig. 5. Moreover, what is the meaning of the black dots? The horizontal bar means the average accuracy, right?

**Other Strengths And Weaknesses:**

Strengths:

1. This paper proposed an efficient tabular foundation model, which shows good balance between performance and speed.

2. The experimental results well support the claims, such as the visualization of feature distribution in Fig. 3 shows that the embeddings are clustered well in a good distribution.

3. The writing is clear and easy to follow.

Weaknesses:

1. In line 240, why should the training be invariant to permutation of columns? If the table is simple one which has n samples on rows and each row has m features, there is no problem. However, if the table becomes more complex like there is a "total (or sum)" column, it will be sensitive to the order of columns, right? A more challenging question is ``can the proposed TabICL handle complex tables and how to handle?"

2. It gives me a contradictory feeling between the feature permutation invariance and mode collapse in the step of row-wise interaction. To avoid mode collapse, the method introduces RoPE. However, it breaks the permutation invariance. Afterwards, to restore invariance, the method ensembles predictions over multiple column permutations. The question is if the features suffer from mode collapse issue again after restoring permutations invariance?

3. Since the performance of the proposed method is so close to TabPFN2, the difference and contributions of the proposed method should be discussed.

4. The comparison of inference speeds with other methods are not listed in paper. The computation cost is essential when deploying a foundation model.

5. The reviewer wonder the concrete examples of the in-context learning in tabular classification. Please list some examples.

    Moreover, can the model handle some real-world tables instead of synthetic ones?

**Questions For Authors:**

please see Strengths And Weaknesses.

**Relation To Broader Scientific Literature:**

This paper may have broader impacts on other disciplinaries as tabular data is used everywhere.

**Theoretical Claims:**

yes

---

> ### Author Rebuttal · Authors · 2025-03-31
>
> We appreciate the reviewer's valuable feedback. We now respond to the reviewer's questions.
>
> ---
> 1. *Why should the training be invariant to permutation of columns?*
>
> In relational data, column order is typically arbitrary. The very visible publication [Why Tabular Foundation Models Should Be a Research Priority] has put forward that tabular methods are expected to be invariant to column order. This is, however, not our battle.
>
> *If there is a 'sum' column, will it be sensitive to column order?*
>
> We may not fully understand the question, but in principle, the position of a "sum" column should not be sensitive to column order.
>
> ---
>
> 2. *It gives me a contradictory feeling ...*
>
> Yes, this is indeed a challenging issue. In fact, both TabPFNv1 and TabPFNv2 face the same mode collapse problem and address it similarly—by first breaking feature-order invariance and then restoring it via ensembling. They differ in how the invariance is broken: TabPFNv2 uses random feature embeddings and feature grouping, while TabPFNv1 uses fixed feature embedding layers.
>
> *Will mode collapse reoccur after restoring permutation invariance?*
>
> No, this will not happen. Ensembling aggregates predictions from independently permuted feature orders. Each permutation receives unique positional encodings via RoPE, leading to diverse, non-collapsed representations. Ensembling effectively averages out positional biases (e.g., effects of column order). This is analogous to data augmentation — diverse permutations provide complementary "views" of the data, enhancing robustness of TabICL. In short, RoPE prevents local collapse (within each permutation), and ensembling restores global invariance without reintroducing collapse. The excellent performance of TabICL also confirms this empirically.
>
> ---
> 3. *Since the performance of the proposed method is so close to TabPFN2...*
>
> While TabICL and TabPFNv2 achieve comparable overall performance, TabICL offers key advantages in handling large datasets, computational efficiency, and scalability — crucial for real-world industrial applications. We acknowledge the need to better clarify these contributions. In the revised manuscript (Related Work, L143–148), we will explicitly emphasize:
> - Methodological differences: TabPFNv2 uses alternating column-row attention, while TabICL adopts a two-stage design (fixed row embeddings + ICL) to reduce complexity.
> - Computational complexity: Detailed comparison is provided in our response to the question 3 of the reviewer rh11.
> - Efficiency and scalability: Including memory optimizations, support for arbitrary class counts, faster inference, etc
> ---
>
> 4. *The comparison of inference speeds with other methods are not listed in paper...*
>
> We focus on inference speed comparison with TabPFNv2, as both are tabular foundation models that perform well without hyperparameter tuning—unlike most other tabular methods. Unfortunately, the TALENT codebase does not report pure inference times excluding tuning and fitting.
>
> When comparing inference speed alone, neither TabPFNv2 nor TabICL shows an advantage. For instance, as noted in the TabPFNv2 paper, on a dataset with 10,000 rows and 10 columns, TabPFNv2 takes 0.2s per sample, while CatBoost (with default settings) requires only 0.0002s for inference.
>
> Their real strength of tabular foundation models lies in the elimination of hyper-parameter tuning and cross-table transferability — qualities that are critical for real-world industrial applications and emphasized in [Why Tabular Foundation Models Should Be a Research Priority].
>
> ---
> 5. *The reviewer wonder the concrete examples of the in-context learning in tabular classification...*
>
> In-context learning for tabular classification can in principle solve all of the applications that regular tabular classification models can solve. For example, some applications of TabPFN are listed at https://github.com/PriorLabs/awesome-tabpfn.
>
> Note that synthetic datasets are only used for pre-training. Yes, our proposed TabICL can deal with real-world datasets. As shown in the paper, TabICL achieves SOTA results on the TALENT benchmark comprising 200 real-world datasets collected from 13 distinct domains, such as education, healthcare, and finance.
>
> ---
> 6. *Suggest to indicate that the red bar ...*
>
> Thank you for your feedback.
>
> - In the caption of figure 5, we will indicate that the red bar represents the median relative accuracy improvements over MLP across all datasets for each method.
>
> - The black dots outside the whiskers represent outliers—datasets where the relative accuracy improvements of a method deviates significantly from the majority of results.
>
> - Horizontal Bars (Whiskers): The horizontal bars (whiskers) extending from the box indicate the range of typical data distribution (approximately 1.5 × the interquartile range from the quartiles). They do not represent average accuracy.

---

> > ### Comment · Reviewer_ba5G · 2025-04-05
> >
> > Thanks for the repsonse. Two concerns still remain.
> >
> > In the first question, I mean "is permutation invariant necessary if the columns are involved with different types of data? For example, 1st-10th column are features, 11th is some other data like the summation of 1st-10th columns."
> >
> > The second concern is that can you list two concrete examples of in-context learning here? For example, list the question to model, and the answer from model.

---

> > > ### Author Response · Authors · 2025-04-07
> > >
> > > Dear reviewer,
> > >
> > > Thank you for your thoughtful follow-up questions. We would like first to present a concrete example to illustrate the internal workflow of TabICL.
> > >
> > > We acknowledge that tabular learning spans a broad range of tasks, including but not limited to Table Question Answering, Retrieval, Generation, and Prediction. TabICL focuses on classification tasks within Table Prediction, where the input is a single table with rows as instances and columns as features.
> > >
> > > The following table shows a concrete example -- Medical Diagnosis Prediction:
> > >
> > >
> > > | Age | Cholesterol | ... | Diagnosis |
> > > | --- | --- | --- | --- |
> > > | 45 | 180 | ... | Healthy |
> > > | 60 | 220 | ... | At Risk |
> > > | 50 | 200 | ... | Healthy |
> > > | ... | ... | ... | ... |
> > > | 55 | 210 | ... | ? |
> > >
> > > TabICL aims to predict the probability of a test sample being healthy given $D_{train}$. More generally, given an input table $X \in \mathbb{R}^{n \times m}$ and a classification target $y$, where $n$ and $m$ denote the number of samples and features respectively, the input-output formulation of TabICL is:
> > >
> > > $p(y_{test}) = \text{TabICL}(x_{test}, D_{train}; \theta)$
> > >
> > > where $\theta$ denotes the parameters of TabICL. Specifically, TabICL processes the data through three sequential transformers: column-wise embedding $TF_{col}$, row-wise interaction $TF_{row}$, and dataset-level in-context prediction $TF_{icl}$.
> > >
> > > Below, we provide a detailed clarification of the two concerns raised.
> > >
> > > ---
> > > 1. *Is permutation invariant necessary...?*
> > >
> > > We clarify that permutation invariance is an assumption — commonly valid in real-world applications where column order carries no semantic meaning. For example, swapping "Age" and "Cholesterol" columns does not alter their semantic roles. Both TabICL and TabPFNv2 are designed under this assumption.
> > >
> > > Of course, permutation invariance may not always hold, especially in the case of spreadsheet formula dependency (formulas may depend on position).
> > > In cases with derived columns (e.g., a “sum” column as the reviewer mentioned), the key is whether TabICL can infer such dependencies through statistical patterns and feature interactions.  This needs to be considered under two scenarios:
> > >
> > > ### a. Position-independent sum column
> > > We assume the sum column carries no additional semantic meaning from its position and is fully computed from other columns: the sum column = sum(all other columns) regardless of column order. In this case,
> > > - For column-wise embedding, $TF_{col}$ encodes each column independently based on its statistical properties. The sum column, probably with distinct statistical patterns (e.g., higher values, lower variance), can be automatically distinguished from raw features by $TF_{col}$.
> > >
> > > - For row-wise interaction, $TF_{row}$ models dependencies between the sum column and contributing features via attention. If the sum column is permuted, this attention should still be able to link it to relevant features based on value patterns instead of position.
> > >
> > > ### b. Position-dependent sum column
> > > We assume the sum column aggregates only preceding columns, then different column permutations yield different values. In this case, we believe TabICL remains effective. As before,
> > >
> > > - Column-wise embedding $TF_{col}$ can still capture the statistical signature of this sum column, regardless of how its values change.
> > > - Meanwhile, row-wise interaction $TF_{row}$ should still be able to model its relationship with relevant columns through attention.
> > >
> > > Of course, there may be other complex scenarios where permutation invariance does not hold. For example, in survey tabular data where each column represents a question, the order of questions may influence responses. Currently, these scenarios where column order explicitly encodes semantics are beyond our current scope but represent an interesting direction for future work.
> > >
> > > To reiterate, TabPFNv2 and TabICL assumes columns are semantically unordered, which holds for most real-world applications.
> > >
> > > ---
> > >
> > > 2. *List two concrete examples ...*
> > >
> > > As mentioned earlier, TabICL is designed for tabular classification tasks, and its input-output formulation is:
> > > $p(y_{test}) = \text{TabICL}(x_{test}, D_{train}; \theta)$
> > >
> > > Beyond the earlier example in Medical Diagnosis Prediction, and as requested by the reviewer, we provide an additional example in Fraud Detection in Transactions:
> > >
> > > | Amount ($) | Merchant Type | Transaction Method | ... | Fraud |
> > > | ---------- | ------------- | ------------------ | --- | ----- |
> > > | 100        | Grocery       | Card Present       | ... | No    |
> > > | 5000       | Electronics   | Online Transfer    | ... | Yes   |
> > > | 200        | Clothing      | Card Present       | ... | No    |
> > > | ...        | ...           | ...                | ... | ...   |
> > >
> > > ---
> > >
> > > Thank you very much for your response. If you have any further questions, please feel free to ask—we’ll be happy to address them as thoroughly as possible.

---

### Official Review · Reviewer_rh11 · 2025-03-14

**Overall Recommendation:** 3

**Summary:**

The paper introduces TabICL, a tabular foundation model designed to scale in-context learning (ICL) for tabular data with a two-stage architecture. This design allows TabICL to handle datasets with up to 500K samples using affordable resources. On the TALENT benchmark with 200 datasets, TabICL achieves comparable performance to TabPFNv2 while being up to ten times faster. On datasets exceeding 10K samples, TabICL outperforms both TabPFNv2 and CatBoost, highlighting its scalability and effectiveness.

**Claims And Evidence:**

The paper supports most claims by empirical results. A few concerns are noted below.

**Essential References Not Discussed:**

The paper covers most works in tabular foundation model learning.

**Experimental Designs Or Analyses:**

In the first step of curriculum training, is the total number of synthetic datasets 100K * 512 = 51M, each with 1024 rows and ≤100 features? Are you using 8 A100 GPUs? The reported training duration (10 days on 8 A100 GPUs) seems longer than expected.

The paper misses ablation studies on key components, such as continual learning and tree-based dataset generation.

Did you compare with the "data grouping based on random forest" method from TabPFNv2, which is used to handle large sample sizes?

**Methods And Evaluation Criteria:**

Why use column-wise embedding and row-wise interaction instead of the opposite?

In Figure 1, why is the notation for the last token 4+m? Since the process involves row-wise interaction, shouldn’t it be 4+n?

What is the overall complexity formulation for the three blocks of TabICL? For the row-wise interaction step, is the time complexity (4+n)^2? If so, how does this remain computationally efficient? Could further formulations on computational efficiency be provided?

For the tree-based synthetic data generator, are you computing the predictions of XGBoost based on its training set? Wouldn't the predicted results be very close to the ground truth on the training set which is essentially random yi? I don't quite get the points here.

**Other Comments Or Suggestions:**

See above.

**Other Strengths And Weaknesses:**

See above.

**Questions For Authors:**

See above.

**Relation To Broader Scientific Literature:**

The paper is well-situated in the context of tabular foundation model learning and ICL.

**Theoretical Claims:**

The paper focuses on improving the empirical performance.

---

> ### Author Rebuttal · Authors · 2025-03-31
>
> We appreciate the reviewer's valuable feedback.
>
> **We kindly invite the reviewer to refer to the figures available at https://limewire.com/d/FhJbU#LPxKd0hlro, which may help better illustrate our responses. The link only contains figures.**
>
> We realize that there may be a misunderstanding of how column-wise embedding and row-wise interaction operate. So, we decide to first explain them.
>
> For tabular data with $n$ rows and $m$ columns:
> - Column-wise embedding applies attention across all $n$ samples within each column, mapping scalar values to embeddings
> - Row-wise interaction applies attention across all $m$ features (plus 4 [CLS] tokens) within each row
>
> We adopted "column-wise"/"row-wise" from BISHOP (ICML 2024), but acknowledge this may cause confusion. Therefore, we will rename:
> - Column-wise embedding → Column-wise inter-sample embedding
> - Row-wise interaction → Row-wise inter-feature interaction
>
> We welcome any alternative naming suggestions from the reviewer.
>
> Next, we respond to your questions.
>
> ---
> 1. *Why use column-wise embedding and row-wise interaction instead of the opposite ?*
> The core idea of TabICL is to encode each row into a fixed-dimensional embedding before applying in-context learning (ICL). Row-wise interaction is used to aggregate row-level information into a single embedding, which positions it between column-wise embedding and ICL.
> ---
> 2. *In Figure 1 ... shouldn’t it be 4+n?*
> As explained above, row-wise inter-feature interaction involves the attention between $m$ features as well as 4 [CLS] tokens within each row. Therefore, it is $4+m$.
> ---
> 3. *What is the overall complexity formulation?*
> Column-wise embedding $TF_{col}$ uses induced self-attention blocks (ISAB), while row-wise interaction $TF_{row}$ and dataset-wise in-context learning $TF_{icl}$ use vanilla self-attention blocks (SAB).
>
> - SAB has quadratic complexity :  $O(L_{seq}^2)$
> - ISAB has linear complexity: $O(L_{seq} \ k)$, where $k$ is the number of inducing vectors
>
> | Block | Description  | Complexity |
> | --- | --- | --- |
> | $\text{TF}_{col}$ | Attn over $n$ samples per column (computed $m$ times)  | $O(n \ k \ m)$ |
> | $\text{TF}_{row}$ | Attn over $m$ features and 4 [CLS] tokens per row (computed $n$ times) | $O((m+4)^2 \ n)$ |
> | $\text{TF}_{icl}$ | Attn over $n$ samples for the full dataset (computed once)  | $O(n^2)$  |
>
> In summary,
> - $TF_{col}$ has linear complexity w.r.t. $n$
> - $TF_{row}$ has quadratic complexity w.r.t. $m$, but $m \ll n$ in most tabular datasets
> - $TF_{icl}$ has quadratic complexity w.r.t. $n$, but it is computed only once because $TF_{row}$ collapses the feature dimension
>
> In contrast, TabPFNv2 uses SAB for sample attention $O(n^2 \ m)$ and feature attention $O(m^2 n)$. Since it never collapses columns, sample attention becomes prohibitive for large $n$ and moderate $m$.
>
> ---
> 4. *For the tree-based synthetic data generator, ... ?*
> We fit a XGBoost regressor to Gaussian noise targets $y_i$. In practice, we limit the max depth to 4 to reduce both fitting time and model capacity, preventing it from fully capturing the randomness. This forces XGBoost to balance between noise and its limited capacity, resulting in a practical randomized initializer—validated by our experimental results.
> ---
> 5. *Are you using 8 A100 GPUs ?*
> We used three A100 GPUs, as mentioned L278-281.
> ---
> 6. *The paper misses ablation studies on key components*
> - Ablation study on curriculum learning
> We evaluate the pre-trained checkpoints after each stage of curriculum learning.  The table below shows the average relative accuracy improvements over the stage 1 (Acc Imp) and the average rank:
>
> |    | Acc Imp across all datasets | Acc Imp across large datasets with > 20K rows | Avg. Rank |
> | --| -- | -- | -- |
> | Stage 1 | -- | -- | 10.52 (7th) |
> | Stage 2 | 2%     | 6%  | 6.87 (2nd)  |
> | Stage 3 | 2.3%  | 7%  | 6.30 (1st)   |
>
> Stages 2 and 3 effectively increase the rank of TabICL from 10.52 (7th) to 6.30 (1st).
> - Ablation study on tree-based generation
> We compare TabICL pre-trained with and without tree-based generation, using the stage 1 setup for 20K steps (to reduce training time). The average relative improvement of tree vs. no tree is as follows:
> | Accuracy | AUC   | Log Loss |
> | --- | --- | --- |
> | 0.37% | 0.18% | 2.5% |
>
> ---
> 7. *Did you compare with the "data grouping based on random forest" method from TabPFNv2?*
> We applied this random forest extension to both TabPFNv2 and TabICL on datasets with over 10K samples (the reported limitation for TabPFNv2). Results show performance gains for both models:
>
> | Model     | Avg. Rank |
> | --- | --- |
> | TabICL    | 6.68 |
> | TabR      | 6.74  |
> | RealMLP   | 6.98 |
> | ModernNCA | 8.52 |
> | TabPFNv2  | 9.14  |
>
> TabPFNv2 improves from 10.89 (see Figure E.5) to 9.14, while TabICL becomes the best-performing method on large datasets !
>
> ---
>
> **In the revised version, we will include complexity analysis, ablation studies, and random forest extension.**

---

### Decision · Program_Chairs · 2025-05-01

**Decision:**

Accept (poster)

**Comment:**

This is clearly a paper whose polarity of views has changed through the review process. The reviewers have particularly appreciated the rebuttals, and have raised their score (rh11, ba5G).

I have read the paper in addition to the rebuttals and exchanges with reviewers. The authors have done a very good job in their rebuttals: lots of insights, new experiments, clarity of explanations. The paper is an experimental paper, which is up to a large extent, not usual in the ML literature about prediction for tabular data where much of the theory developed in ML applies without hurdles, especially for models like decision trees.

Yet, the paper manages to develop in a clear way its experiments and it makes little doubt that, *given the inclusion of the experiments promised to the reviewers* (ablation studies, and random forest extension for rh11, tabM for gpuG, LocalPFN and TuneTables for VxC8) but also the additional material produced at rebuttal time (complexity analysis rh11, additional explanations ba5G), the paper may end up being a sound benchmark for DL on tabular data.

It should be accepted.